# Ethanolaminephosphate cytidylyltransferase is essential for survival, lipid homeostasis and stress tolerance in *Leishmania major*

**Somrita Basu[1], Mattie C. Pawlowic[1,2], Fong-Fu Hsu[3], Geoff Thomas[1], Kai Zhang[1]***

**1** Department of Biological Sciences, Texas Tech University, Lubbock, Texas, United States of America,
**2** Wellcome Centre for Anti-Infectives Research (WCAIR), Division of Biological Chemistry and Drug Discovery, School of Life Sciences, University of Dundee, Dundee, United Kingdom, **3** Mass Spectrometry Resource, Division of Endocrinology, Metabolism, and Lipid Research, Department of Medicine, Washington University School of Medicine, Saint Louis, Missouri, United States of America

\* kai.zhang@ttu.edu.

**Data Availability Statement:** All relevant data are within the manuscript and its Supporting Information files.

## Abstract

Glycerophospholipids including phosphatidylethanolamine (PE) and phosphatidylcholine (PC) are vital components of biological membranes. Trypanosomatid parasites of the genus *Leishmania* can acquire PE and PC via *de novo* synthesis and the uptake/remodeling of host lipids. In this study, we investigated the ethanolaminephosphate cytidylyltransferase (EPCT) in *Leishmania major*, which is the causative agent for cutaneous leishmaniasis. EPCT is a key enzyme in the ethanolamine branch of the Kennedy pathway which is responsible for the *de novo* synthesis of PE. Our results demonstrate that *L. major* EPCT is a cytosolic protein capable of catalyzing the formation of CDP-ethanolamine from ethanolaminephosphate and cytidine triphosphate. Genetic manipulation experiments indicate that EPCT is essential in both the promastigote and amastigote stages of *L. major* as the chromosomal null mutants cannot survive without the episomal expression of EPCT. This differs from our previous findings on the choline branch of the Kennedy pathway (responsible for PC synthesis) which is required only in promastigotes but not amastigotes. While episomal EPCT expression does not affect promastigote proliferation under normal conditions, it leads to reduced production of ethanolamine plasmalogen or plasmenylethanolamine, the dominant PE subtype in *Leishmania*. In addition, parasites with episomal EPCT exhibit heightened sensitivity to acidic pH and starvation stress, and significant reduction in virulence. In summary, our investigation demonstrates that proper regulation of EPCT expression is crucial for PE synthesis, stress response, and survival of *Leishmania* parasites throughout their life cycle.

## Author summary

In nature, *Leishmania* parasites alternate between fast replicating, extracellular promastigotes in sand fly gut and slow growing, intracellular amastigotes in macrophages. Previous studies suggest that promastigotes acquire most of their lipids via *de novo* synthesis

**Funding:** This work was supported by the National Institutes of Health (R15AI156746 and R01AI139198 to KZ, P41-GM103422, P60-DK20579 and P30-DK56341 to FH and the Biomedical Mass Spectrometry Resource at Washington University in St. Louis) and Howard Hughes Medical Institute (Undergraduate Science Education Program to MP). The funders have no role in the study design, data collection and analysis, decision to publish, or preparation of the manuscript.

**Competing interests:** The authors have declared that no competing interests exist.

whereas amastigotes rely on the uptake and remodeling of host lipids. Here we investigated the function of ethanolaminephosphate cytidyltransferase (EPCT) which catalyzes a key step in the *de novo* synthesis of phosphatidylethanolamine (PE) in *Leishmania major*. Results showed that EPCT is indispensable for both promastigotes and amastigotes, indicating that *de novo* PE synthesis is still needed at certain capacity for the intracellular form of *Leishmania* parasites. In addition, elevated EPCT expression alters overall PE synthesis and compromises parasite's tolerance to adverse conditions and is deleterious to the growth of intracellular amastigotes. These findings provide new insight into how *Leishmania* acquire essential phospholipids and how disturbance of lipid metabolism can impact parasite fitness.

## Introduction

Protozoan parasites of the genus *Leishmania* are transmitted through the bite of hematophagous sand flies. During their life cycle, *Leishmania* parasites alternate between flagellated, extracellular promastigotes in sand fly midgut and non-flagellated, intracellular amastigotes in mammalian macrophages. These parasites cause leishmaniasis which ranks among the top ten neglected tropical diseases with 10–12 million people infected and 350 million people at the risk of acquiring infection [1,2]. Drugs for leishmaniasis are plagued with strong toxicity, low efficacy, and high cost [3,4]. To develop better treatment, it is necessary to gain insights into how *Leishmania* acquire essential nutrients and proliferate in the harsh environment in the vector host and mammalian host.

To sustain growth, *Leishmania* parasites must generate abundant amounts of lipids including glycerophospholipids, sterols and sphingolipids. Phosphatidylethanolamine (PE) and phosphatidylcholine (PC) are two common classes of glycerophospholipids. Besides being major membrane constituents, PE and PC can function as precursors for several signaling molecules and metabolic intermediates including lyso-phospholipids, phosphatidic acid, diacylglycerol, and free fatty acids [5,6]. In addition, PE contributes to the synthesis of GPI-anchored proteins in trypanosomatid parasites by providing the ethanolamine phosphate bridge that links proteins to glycan anchors [7]. PE is also involved in the formation of autophagosome during differentiation and starvation in *Leishmania major* [8,9] and the posttranslational modification of eukaryotic elongation factor 1A in *T. brucei* [10].

For many eukaryotic cells, the majority of PE and PC are synthesized *de novo* via the Kennedy pathway (Fig 1) [11]. In *Leishmania*, the key metabolite ethanolamine phosphate (EtN-P) is generated from of the sphingoid base metabolism or the phosphorylation of ethanolamine (EtN) by ethanolamine kinase [12](Fig 1). EtN-P is then conjugated to cytidine triphosphate (CTP) to produce CDP-EtN and pyrophosphate by the enzyme ethanolaminephosphate cytidylyltransferase (EPCT) [13]. In the EtN branch of the Kennedy pathway, through the activity of ethanolamine phosphotransferase (EPT), CDP-EtN is combined with 1-alkyl-2-acyl-glycerol to eventually generate plasmenylethanolamine (PME) or ethanolamine plasmalogen, the dominant subtype of PE in *Leishmania* [14]. CDP-EtN can also be combined with 1,2-diacyl-glycerol to form 1,2-diacyl-PE (a minor subtype of PE in *Leishmania*) by choline ethanolamine phosphotransferase (CEPT) [15,16]. A similar branch of the Kennedy pathway is responsible for the *de novo* synthesis of PC (choline → choline-phosphate → CDP-choline → PC), with CEPT catalyzing the last step of conjugating CDP-choline and diacylglycerol into PC as a dual activity enzyme [16–18]. Besides the Kennedy pathway, PE may be generated from the

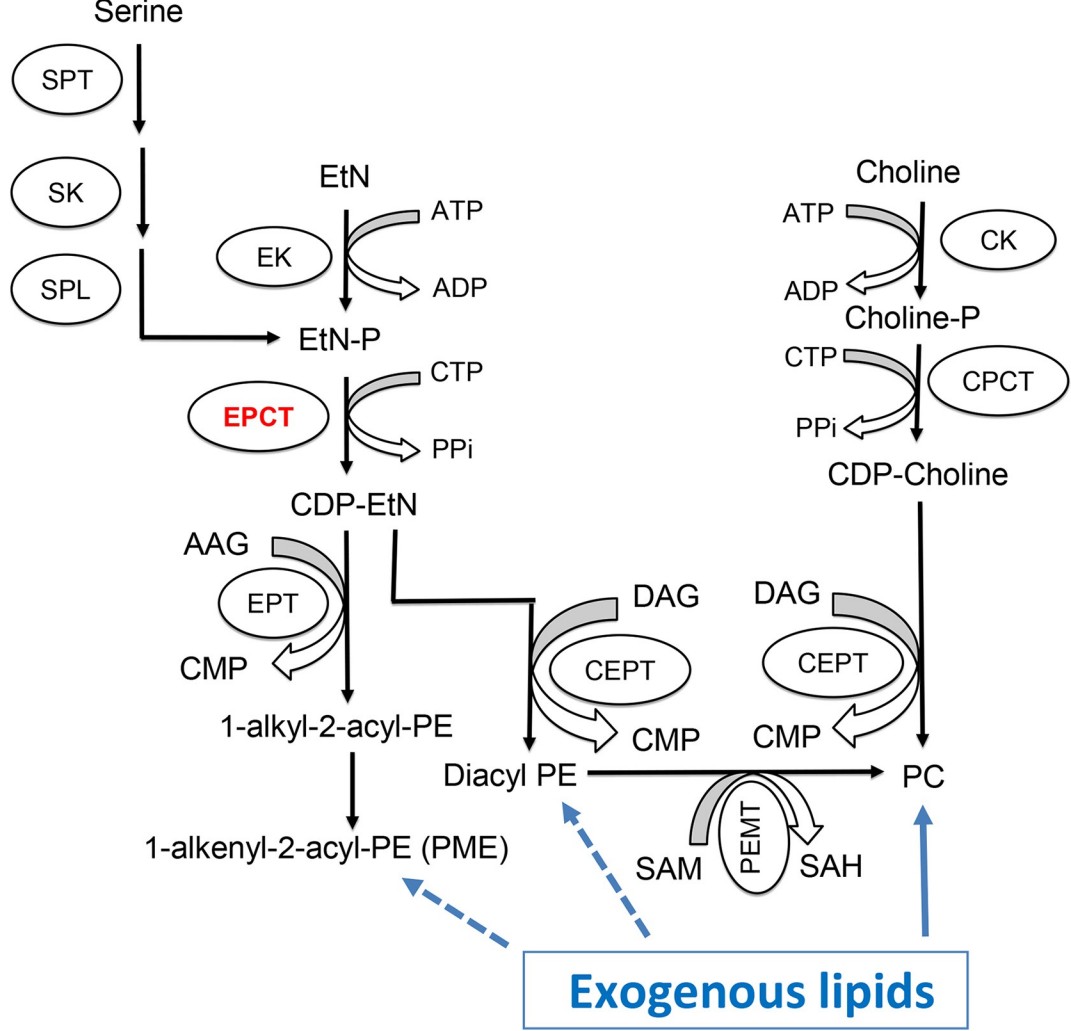

**Fig 1. Synthesis of PE and PC in *Leishmania*.** SPT: Serine palmitoyltransferase; SK: Sphingosine kinase; SPL: sphingosine-1-phosphate lyase; EK: ethanolamine kinase; EPCT: ethanolaminephosphate cytidylyltransferase; EPT: ethanolamine phosphotransferase; CK: choline kinase; CPCT: cholinephosphate cytidylyltransferase; CEPT: choline ethanolamine phosphotransferase; PEMT: Phosphatidylethanolamine N-methyltransferase; EtN: ethanolamine; EtN-P: ethanolamine phosphate; AAG: 1-alkyl2-acyl glycerol; DAG: 1,2-diacylglycerol; CTP: cytidine triphosphate; CDP: cytidine diphosphate; CMP: cytidine monophosphate; PPi: pyrophosphate; PME: plasmenylethanolamine; diacyl PE: 1,2-diacyl-phosphatidylethanolamine, PC: 1,2-diacyl-phosphatidylcholine; SAM: S-adenosyl-L-methionine; SAH: S-adenosyl-L-homocysteine.

decarboxylation of phosphatidylserine (PS) in the mitochondria and PC may be generated from the N-methylation of PE [19–21].

In addition to biosynthesis, *Leishmania* parasites (especially the intracellular amastigotes) can take up and modify existing lipids to fulfill their needs [22–24]. For example, the enzyme CEPT which is directly responsible for producing PC and diacyl-PE (Fig 1) is essential for *L. major* promastigotes in culture but is not required for the survival or proliferation of amastigotes in mice [18]. While diacyl-PE is a minor lipid component in *Leishmania*, PC is highly abundant in both promastigotes and amastigotes [25,26]. These results argue that *de novo* PC synthesis is required to generate large amount of lipids to support rapid parasite replication during the promastigote stage (estimated doubling time: 6–8 hours in culture and 10–12 hours

in sand fly) [27,28]. In contrast, intracellular amastigotes can acquire enough PC through the uptake and remodeling of host lipids, which seems to fit their slow growing, metabolically quiescent state (estimated doubling time: 60 hours) [29,30]. Consistent with these findings, sphingolipid analyses of *L. major* amastigotes revealed high levels of sphingomyelin which could not be synthesized by *Leishmania* but was plentiful in mammalian cells [31]. Conversely, the biosynthesis of parasite-specific sphingolipid, inositol phosphorylceramide (IPC), is fully dispensable for *L. major* amastigotes [12,31,32]. Collectively, these data suggest that as *Leishmania* transition from fast-replicating promastigotes to slow-growing amastigotes, they undergo a metabolic switch from *de novo* synthesis to scavenge/remodeling to acquire their lipids [24].

Despite of these findings, it is important to explore whether intracellular amastigotes retain some capacity for *de novo* lipid synthesis. Our previous studies on the sterol biosynthetic mutant *c14dm̄* suggest that is the case [33]. *L. major c14dm̄* mutants lack the sterol-14α-demethylase which catalyzes the removal of C-14 methyl group from sterol intermediates. Promastigotes of *c14dm̄* are devoid of ergostane-type sterols (which are abundant in wild type [WT] parasites) and accumulate 14-methyl sterol intermediates. However, lipid analyses of lesion-derived amastigotes demonstrate that both WT and *c14dm̄* amastigotes contain cholesterol (which must be taken from the host since *Leishmania* only synthesize ergostane-type sterols) as their main sterol [33]. Parasite-specific sterols, i.e., ergostane-type sterols such as ergosterol and 5-dehydroepisterol in WT amastigotes and C-14-methylated sterols in *c14dm̄* amastigotes are only detected at trace levels. Nonetheless, *c14dm̄* amastigotes show significantly attenuated virulence and reduced growth in comparison to WT and *C14DM* add-back amastigotes, suggesting that the residual amounts of endogenous sterols (which cannot be scavenged from the host) play pivotal roles in amastigotes which may constitute the basis for sterol biosynthetic inhibitors as anti-leishmaniasis drugs [33,34].

In this study, we investigated the roles of EPCT (EC 2.7.7.14) which catalyzes the conversion of EtN-P and CTP into CDP-EtN and pyrophosphate in the Kennedy pathway (Fig 1). Our previous study on EPT in *L. major* revealed that EPT is mostly required for the synthesis of PME but not diacyl PE or PC [14]. Disruption of EPCT, on the other hand, may have a much more profound impact on the synthesis of PME, diacyl PE and PC (which can be generated from diacyl PE via N-methylation) (Fig 1). In mammalian cells, EPCT is considered as the rate limiting enzyme in the EtN branch of the Kennedy pathway and a key regulator of PE synthesis [35,36]. Disruption of EPCT in mice leads to developmental defects and embryonic lethality [37,38]. Additional studies report EPCT heterozygous mice have increased diacylglycerol and triacylglycerol in liver that resemble features of metabolic diseases, suggesting that EPCT is involved in maintaining the homeostasis of neutral lipids as well [38,39]. In *Trypanosoma brucei* (a related trypanosomatid protozoan), RNAi knockdown of EPCT reduces *de novo* PE synthesis leading to growth arrest and altered mitochondrial morphology in procyclics [16,40,41]. Based on these findings, it is of interest to determine the roles of EPCT in *Leishmania* especially during the disease-causing intracellular stage when parasites acquire most of their lipids via scavenging. Our results indicate that EPCT is essential for the survival of both promastigotes and amastigotes in *L. major*. In addition, the expression level of EPCT is crucial for stress tolerance, lipid homeostasis and the synthesis of GPI-anchored proteins. These findings may inform the development of new anti-*Leishmania* drugs.

## Results

### Identification and functional verification of *L. major* EPCT

A putative *EPCT* gene was identified in the genome of *L. major* (Tritrypdb: LmjF 32.0890) with synthetic orthologs in other *Leishmania* spp., *Trypanosoma brucei*, and *Trypanosoma*

*cruzi*. The predicted *L. major* EPCT protein consists of 402 amino acids and bears 35–42% identity to EPCTs from *Saccharomyces cerevisiae*, *Plasmodium falciparum*, *Arabidopsis thaliana*, and *Homo sapiens* (S1 Fig).

To confirm its function, the *EPCT* open reading frame (ORF) was cloned into a pXG plasmid (a high copy number protein expression vector) [42] and introduced into *L. major* wild type parasites (WT) as WT+EPCT. When promastigote lysate from WT+EPCT was incubated with [$^{14}$C]-EtN-P and CTP for 20 minutes at room temperature, we detected a radiolabeled product which exhibited similar mobility (retention factor) as pure CDP-EtN on thin layer chromatography (TLC), suggesting it was [$^{14}$C]-CDP-EtN (Figs 2A and S2A). Radiolabeled CDP-EtN was also formed when [$^{14}$C]-EtN-P and CTP were incubated with mouse liver homogenate, but not boiled lysates (Figs 2A and S2B). We did not detect significant amount of [$^{14}$C]-CDP-EtN using WT lysate, indicating that the basal level of EPCT activity in *L. major* promastigotes was below the level of detection with this approach (Fig 2A and 2B, activity from WT was similar to buffer alone and boiled lysate). To determine the cellular localization of EPCT, a GFP-tagged EPCT was cloned into pXG plasmid and introduced into WT promastigotes to generate WT+GFP-EPCT. These WT+GFP-EPCT cells displayed similar EPCT activity levels as WT+EPCT cells (Fig 2A and 2B; 11–12 times higher than WT or boiled lysate, $p < 0.001$). By western blot, GFP-EPCT was detected at the predicted molecular weight of ~73 kDa (Fig 2C) and mainly found in the cytosolic fraction after sub-cellular fractionation (Fig 2D). A previously reported cytoplasmic protein HSP83 [43] was detected in both the cytosolic and the large granule fractions (pellet 10K), whereas the plasma membrane-localized lipophosphoglycan (LPG) [44] was mainly found in the large granule fraction (Fig 2D). In addition, GFP-EPCT exhibited a cytoplasmic localization by fluorescence microscopy (Fig 2E). Together, these findings argue that *L. major* EPCT is a cytosolic enzyme capable of condensing EtN-P and CTP into CDP-EtN (Fig 1).

## Chromosomal *EPCT*-null mutants cannot be generated without a complementing episome

To determine whether EPCT is required for survival in *L. major*, we first attempted to delete the two chromosomal *EPCT* alleles using the homologous recombination approach [45]. As demonstrated by Southern blot (S3A and S3B Fig), we successfully replaced one *EPCT* allele with the blasticidin resistance gene (*BSD*) using this approach and generated several heterozygous *EPCT*+/- clones. However, repeated attempts to delete the second *EPCT* allele failed to recover any true knockouts (results from one attempt were shown in S3C and S3D Fig), suggesting that a different method is needed to generate chromosomal *EPCT*-null mutants.

We then adopted the complementing episome-assisted knockout approach that has been used to study essential genes in *Leishmania* [46–48]. To do so, a pXNG4-EPCT plasmid (complementing episome) containing genes for *EPCT*, green fluorescence protein (*GFP*), nourseothricin resistance (*SAT*) and thymidine kinase (*TK*) was constructed and introduced into *EPCT*+/- parasites (*EPCT*+/- +pXNG4-EPCT), followed by attempts to delete the second chromosomal *EPCT* allele with the puromycin resistance gene (*PAC*) (S4 Fig). With this approach, we were able to generate multiple clones showing successful replacement of chromosomal *EPCT* with *BSD* and *PAC* (*epct*‾+ pXNG4-EPCT in Figs 3A–3D and S4). Southern blot with an *EPCT* ORF probe revealed high levels of pXNG4-EPCT plasmid (20–35 copies/cell) in these *epct*‾+ pXNG4-EPCT parasites but no endogenous *EPCT* was detected (Figs 3C, 3D and S4). Overexpression of *EPCT* from the pXNG4-EPCT plasmid did not have any significant impact on promastigote replication under normal culture conditions (Fig 3E).

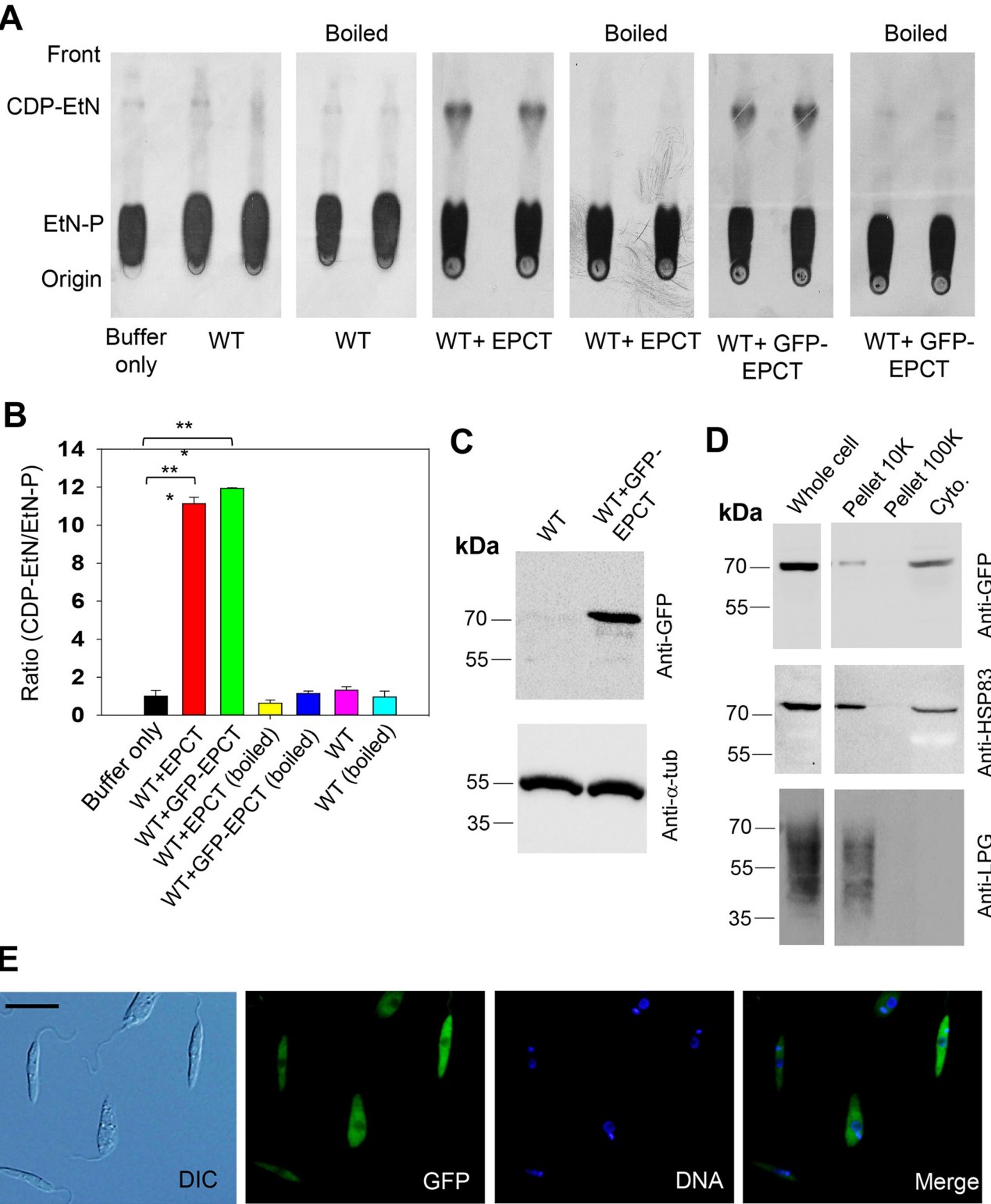

**Fig 2. *L. major* EPCT is a functional enzyme located in the cytoplasm.** (**A**-**B**) Whole cell lysates (boiled and not boiled, two repeats were shown) of WT, WT+EPCT, and WT+GFP-EPCT promastigotes were incubated with [$^{14}$C]-EtN-P followed by TLC analysis as described in *Materials and Methods*. Representative TLC images were shown in **A** and the average ratios of [$^{14}$C]-CDP-EtN:[$^{14}$C]-EtN-P were normalized to WT (as 1.0) and summarized in **B** (error bars represent standard deviations from four repeats, ***: $p < 0.001$). (**C**) Promastigote lysates of WT and WT + GFP-EPCT were analyzed by western-blot using an anti-GFP antibody (top) or anti-α-tubulin antibody (bottom)(**D**) Whole cell lysate (1.0 x 10$^6$ cells) and sub-

cellular fractions of WT + GFP-EPCT cells were probed by anti-GFP (top), -HSP83 (middle), or -LPG (bottom) antibodies. (**E**) Log phase promastigotes of WT+GFP-EPCT were examined by fluorescence microscopy. DIC: differential interference contrast. Scale bar: 10 μm. DNA: staining with Hoechst 33342. Merge: overlay of GFP and DNA.

### EPCT is essential for *L. major* promastigotes

To evaluate the essentiality of EPCT, *EPCT+/-* +pXNG4-EPCT and *epct̄*+ pXNG4-EPCT promastigotes were cultivated in the presence or absence of ganciclovir (GCV). Because of the *TK* expression from pXNG4-EPCT, adding GCV would trigger premature termination of DNA synthesis [46]. Thus, in the presence of GCV, parasites would favor the elimination of pXNG4-EPCT (which can be tracked by monitoring GFP fluorescence level) during replication to avoid toxicity, if the episome is dispensable [46]. As shown in Fig 4A, after being cultivated in the presence of GCV for 14 consecutive passages, those chromosomal *EPCT*-null promastigotes still contained 60–74% of GFP-high cells, indicative of high pXNG4-EPCT retention levels after prolonged negative selection. In contrast, the pXNG4-EPCT plasmid was easily expelled from the *EPCT+/-* +pXNG4-EPCT parasites after 8 passages in GCV (their GFP level dropped to that of WT, Fig 4A and 4B), suggesting that the remaining chromosomal *EPCT* allele made the episome expendable. Without GCV, we detected a gradual loss of episome in the *EPCT+/-* +pXNG4-EPCT but not *epct̄*+ pXNG4-*EPCT* parasites (Fig 4A), again arguing that the episome is indispensable in those chromosomal-null mutants. As controls, *EPCT+/-* +pXNG4-EPCT and *epct̄*+ pXNG4-EPCT grown in the presence of nourseothricin (the positive selection) consistently retained 85–91% GFP-high cells ("+SAT" in Fig 4A and 4C).

Our analysis showed that 25–38% of *epct̄*+ pXNG4-EPCT parasites became GFP-low after prolonged exposure to GCV (Fig 4A and 4D). To examine whether these GFP-low cells in *epct̄* + pXNG4-EPCT could survive by themselves, they were separated from GFP-high cells by fluorescence-activated cell sorting (FACS) and individual clones were isolated after serial dilution. As illustrated in S5 Fig, when single clones were expanded from the sorted GFP-low population in the presence of GCV and absence of nourseothricin, they quickly regained GFP fluorescence. After two passages, two such clones showed similar percentages of GFP-high cells (70–74%) as the population prior to sorting (Figs 4D–4F and S5). These data suggest that GFP-low cells still contained a low level of episome and would increase the episome level during proliferation after they were separated from GFP-high cells. Additionally, quantitative PCR (qPCR) analyses were performed on these clones using primers targeting the *GFP* region (to determine total plasmid copy number) and the *L. major* 28S rDNA (to determine total parasite number). Results revealed ~8 copies of pXNG4-EPCT plasmid per cell in the *epct̄* + pXNG4-EPCT GCV-treated clones, whereas the *EPCT+/-* +pXNG4-EPCT clones isolated by the same process (GCV treatment ➔ GFP-low sorting ➔ serial dilution) contained <0.01 copy per cell (S6A Fig). We also sequenced the pXNG4-EPCT plasmid DNA from GCV-treated *epct̄* + pXNG4-EPCT clones and did not find mutations in *TK*, *GFP* or *EPCT*. Finally, we found a significant growth delay in GCV-treated *epct̄*+ pXNG4-EPCT but not *EPCT+/-* +- pXNG4-EPCT promastigotes in comparison to control cells (S6B Fig). This is consistent with the episome retention in *epct̄*+ pXNG4-EPCT which leads to GCV-induced cytotoxicity. Together, these findings demonstrate the essential nature of EPCT in *L. major* promastigotes.

### EPCT expression is crucial for *L. major* amastigotes

To investigate if EPCT is required for the survival of intracellular amastigotes, we infected BALB/c mice in the footpad with stationary phase promastigotes. After infection, half of the mice received daily GCV treatment, and the other half received equivalent amount of PBS

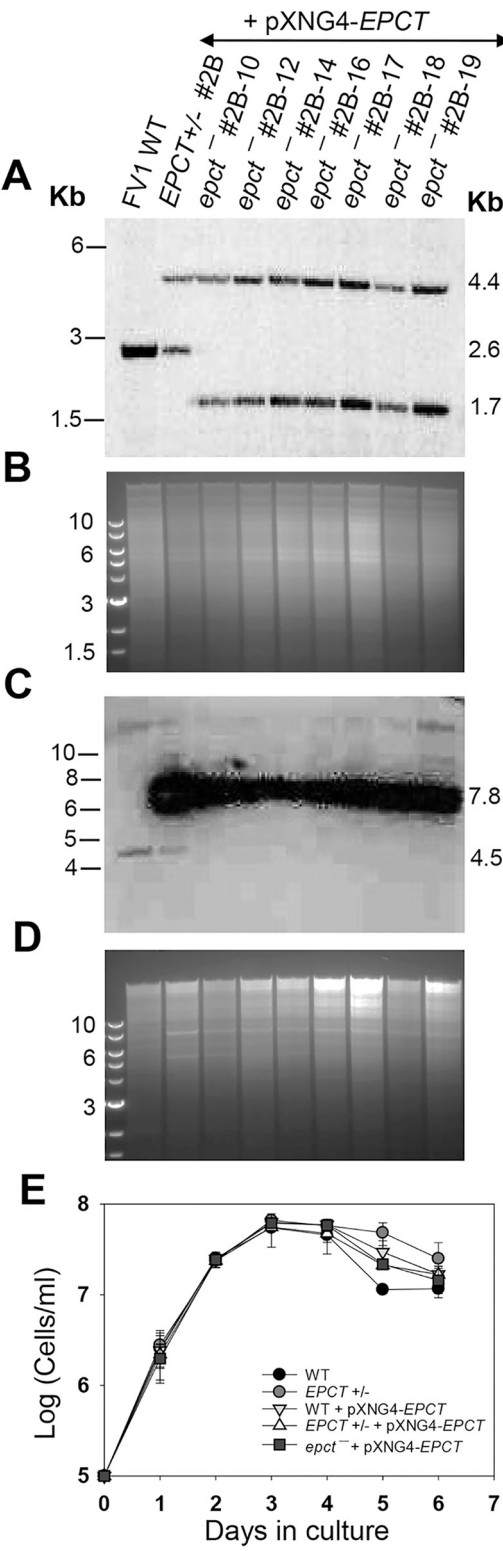

**Fig 3. Generation of chromosomal *EPCT*-null mutants using an episome-assisted approach.** (**A**-**D**) Genomic DNA samples from FV1 WT, *EPCT*+/- + pXNG4-*EPCT* (clone #2B), and *epct⁻* + pXNG4-*EPCT* (seven clones) parasites were digested with *Aat* II (**A**, **B**) or *Kpn* I+*Avr* II (**C**, **D**) and analyzed by Southern blot using radiolabeled probes for an upstream flanking sequence (**A**, **B**) or the open reading frame of *EPCT* (**C**, **D**). DNA loading controls with ethidium bromide staining for **A** and **C** were included in **B** and **D**, respectively. The scheme of Southern blot and expected DNA

fragment sizes were shown in S3 Fig. (**E**) Promastigotes of WT, *EPCT*+/- (#2), WT + pXNG4-*EPCT*, *EPCT*+/−
+ pXNG4-*EPCT* (clone #2B), and *epct* +pXNG4-*EPCT* (clone #2B-10) were cultivated at 27°C in complete M199
media and culture densities were determined daily using a hemocytometer. Error bars indicate standard deviations
from three biological repeats.

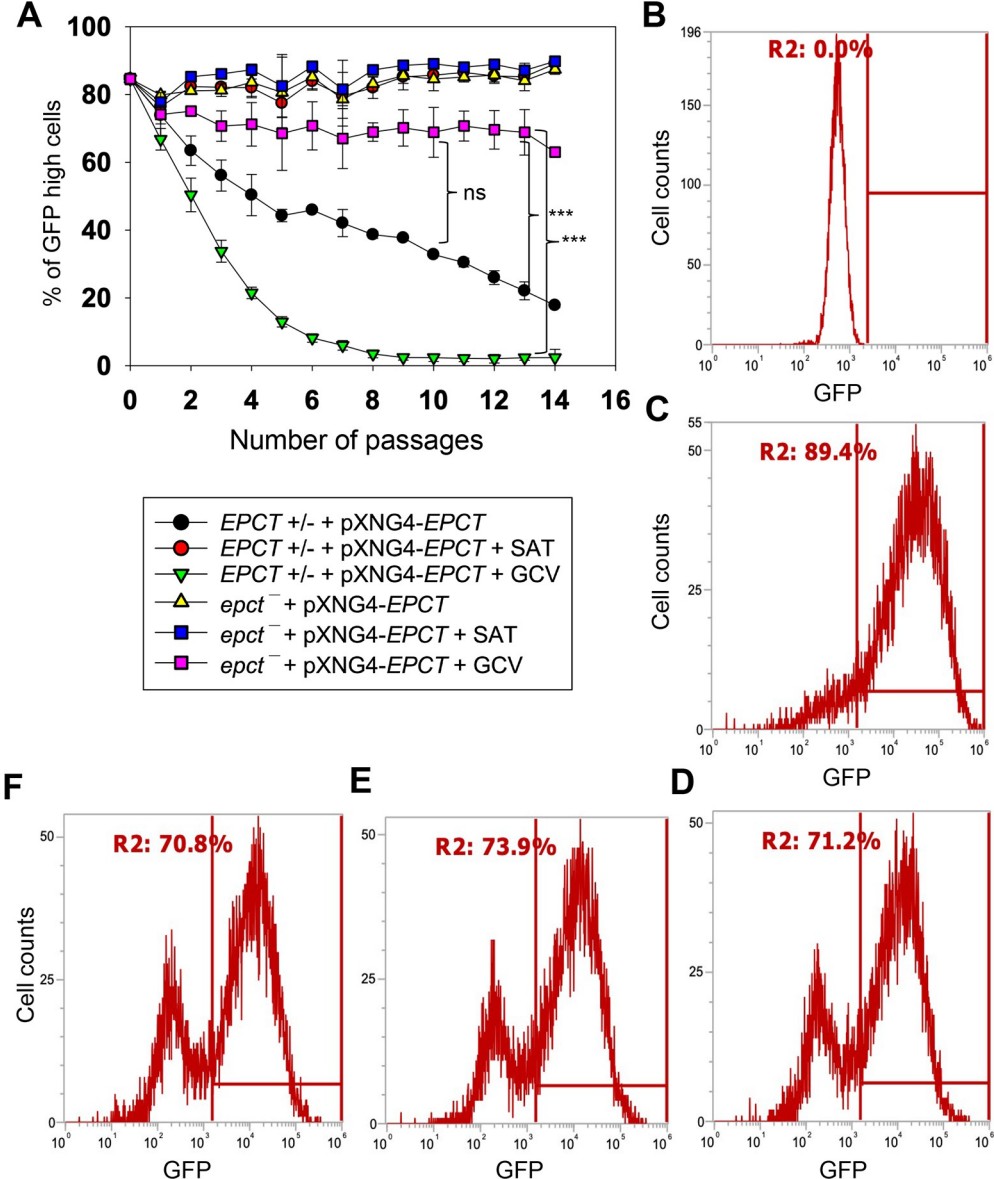

**Fig 4. EPCT is indispensable during the promastigote stage.** (**A**) Promastigotes were continuously cultivated in the
presence or absence of ganciclovir (GCV) or nourseothricin (SAT) and passed every three days. Percentages of GFP-
high cells were determined for every passage. Error bars represent standard deviations from three repeats (**: $p < 0.01$,
***: $p < 0.001$). (**B-D**) After 14 passages, WT (**B**) and *epct*+pXNG4-EPCT parasites grown in the presence of SAT (**C**)
or GCV (**D**) were analyzed by flow cytometry to determine the percentages of GFP-high cells (indicated by R2 in the
histograms). (**E-F**) Two clones were isolated from the GFP-low population in **D** by FACS, amplified in the absence of
SAT and presence of GCV, and analyzed for GFP expression levels by flow cytometry (more details for the sorting
were included in S5 Fig).

(control group) for 14 consecutive days as previously described [48]. No significant changes in mouse body weight or movement were detected from GCV treatment (S7A Fig). Mice infected by WT and *EPCT+/-* parasites exhibited equally rapid development of footpad lesions that correlated with robust parasite replication, indicating that one chromosomal copy of *EPCT* is sufficient for the mammalian stage (Fig 5A and 5B). By comparison, mice infected by *EPCT+/-* +pXNG4-EPCT or *epct*‾+ pXNG4-EPCT showed a 9-12-week delay in lesion development (Fig 5A), and the delay was consistent with the significantly reduced parasite growth as determined by limiting dilution assay and qPCR (Figs 5B and S7B). These results suggest that elevated *EPCT* expression from pXNG4-EPCT, a high copy number plasmid (Figs 3C and 5C), may be responsible for the delayed lesion progression and amastigote growth from *EPCT+/-* +-pXNG4-EPCT and *epct*‾+ pXNG4-EPCT promastigotes (Figs 5 and S7). To test this hypothesis, we introduced pXNG4-EPCT into WT parasites and the resulting WT +pXNG4-EPCT cells showed similar infectivity in mice as *EPCT+/-* +pXNG4-EPCT or *epct*‾+ pXNG4-EPCT parasites (Fig 5), indicating that this plasmid can attenuate virulence in the WT background as well. In addition, the *EPCT* gene was cloned into a pGEM vector along with its 5'- and 3'-flanking sequences (~1 Kb each) and the resulting pGEM-EPCT was introduced into *EPCT+/-* parasites. Like cells with pXNG4-EPCT, these *EPCT+/-* + pGEM-EPCT parasites displayed significantly reduced virulence and growth in BALB/c mice (S8 Fig), proving that these defects were caused by *EPCT* overexpression and not restricted to the pXNG4 plasmid.

While GCV treatment had little impact on infections caused by WT, *EPCT+/-*, or *EPCT+/-* +pXNG4-EPCT parasites, it caused an additional 3-4-week delay in lesion progression for *epct*‾ + pXNG4-EPCT (Fig 5A and 5B), suggesting that GCV-induced cytotoxicity is more significant in the chromosomal null mutants than others. To determine the pXNG4-EPCT copy number in amastigotes, we performed qPCR analyses on genomic DNA extracted from lesion-derived amastigotes. The average plasmid copy number per cell was revealed by dividing the total plasmid copy number with total parasite number. As shown in Fig 5C, *EPCT+/-* +-pXNG4-EPCT amastigotes contained 8–13 copies per cell at week 4 post infection and that number gradually went down to 1.5–2.5 copies per cell by week 14–20; and GCV treatment caused reduction in plasmid copy number as expected. The reduction in plasmid copy number over time suggests that the episome is not required in *EPCT+/-* +pXNG4-EPCT amastigotes. Similarly, the episome copy number in WT +pXNG4-EPCT amastigotes decreased from 14–16 per cell at week 4 post infection to 2–4 per cell in week 14–20 and GCV treatment led to further reduction (Fig 5C). In comparison, *epct*‾+ pXNG4-EPCT amastigotes maintained 12–24 plasmid copies per cell throughout the course of infection even with GCV treatment (Fig 5C). Thus, these chromosomal *EPCT*-null amastigotes must retain a high episome level to survive which is consistent with their increased sensitivity to GCV (Fig 5A and 5B), Together, these data argue that EPCT is critically important for the survival of intracellular amastigotes.

In addition to plasmid copy number, we also examined EPCT transcript levels by reverse transcription-qPCR using α-tubulin transcript as the internal standard (Fig 6). For WT parasites, EPCT mRNA level went down ~60% from promastigotes to amastigotes, consistent with the relatively slow replication rate for amastigotes (Fig 6A). As expected, the presence of pXNG4-EPCT resulted in a 10-18-fold increase in EPCT mRNA levels (compared to WT cells) during the promastigote stage (Fig 6B). While the overexpression was less pronounced during the mammalian stage, we still observed a 2-8-fold increase in EPCT mRNA in *epct*‾ + pXNG4-EPCT amastigotes (week 14–20 post infection) in comparison to WT amastigotes (Fig 6C). For *EPCT+/-* +pXNG4-EPCT and *epct*‾+ pXNG4-EPCT amastigotes (week 14 post infection), EPCT transcript levels correlated with plasmid copy numbers (Figs 5C and 6C). The fact that GCV treatment increased EPCT levels in *epct*‾+ pXNG4-EPCT may be due to a combination of factors including the level of GCV being below the minimum effective

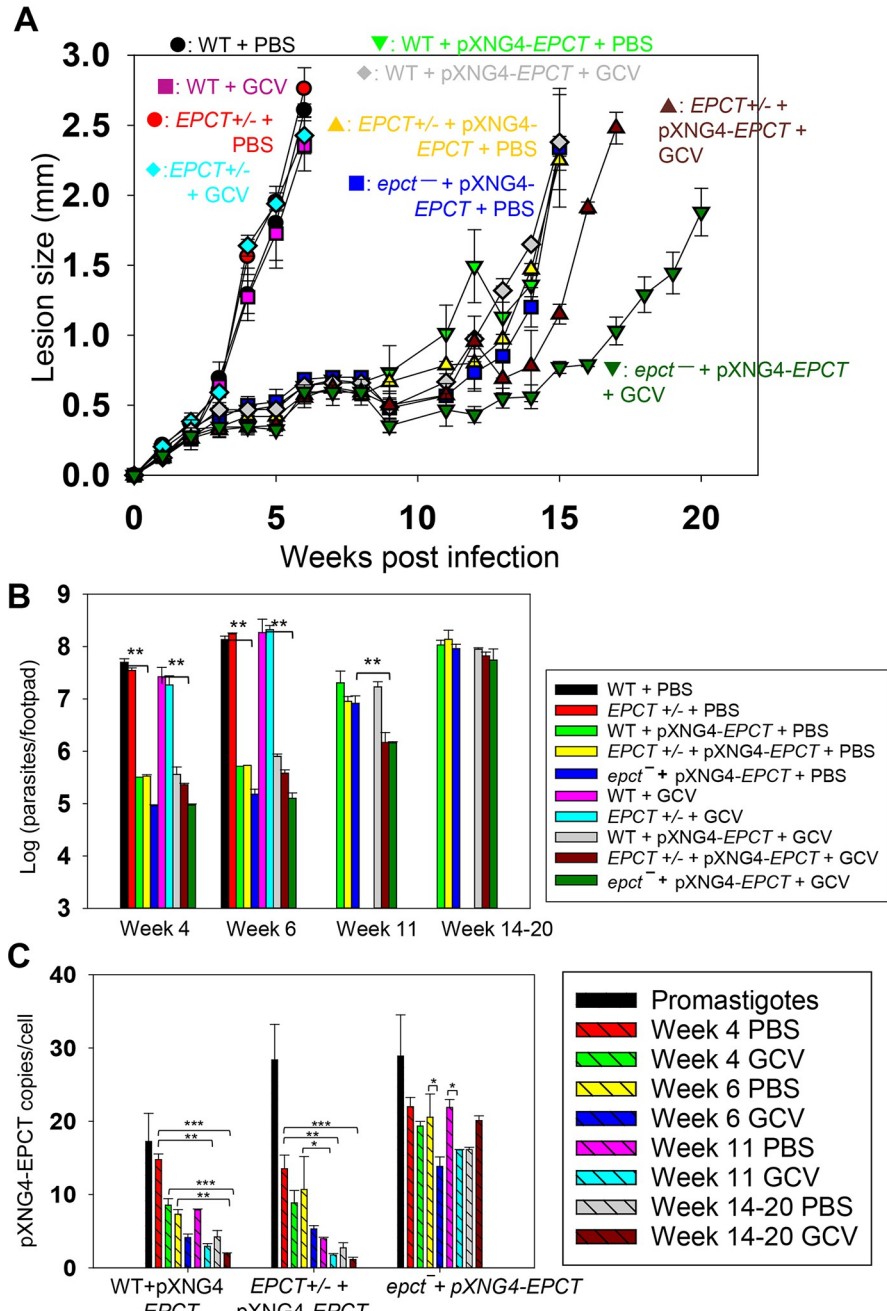

**Fig 5. EPCT is indispensable during the amastigote stage.** BALB/c mice were infected in the footpad with stationary phase promastigotes and treated with GCV or PBS as described in *Materials and Methods*. (**A**) Sizes of footpad lesions for infected mice were measured weekly. (**B**) Parasite numbers in infected footpads were determined by limiting dilution assay at the indicated times. Data for WT and *EPCT+/-* amastigotes were not available after 8 weeks when those infected mice had reached the humane endpoint. (**C**) The average pXNG4-EPCT plasmid copy numbers in promastigotes and amastigotes (#/cell ± SDs) were determined by qPCR. Error bars represent standard deviations from three repeats (*: $p < 0.05$, **: $p < 0.01$, ***: $p < 0.001$).

concentration and amastigotes starting rapid replication by week 14. For WT +pXNG4-EPCT amastigotes (week 14 post infection), GCV treatment had opposite effects on EPCT transcription (upregulation) and plasmid copy number (downregulation), which may reflect increased

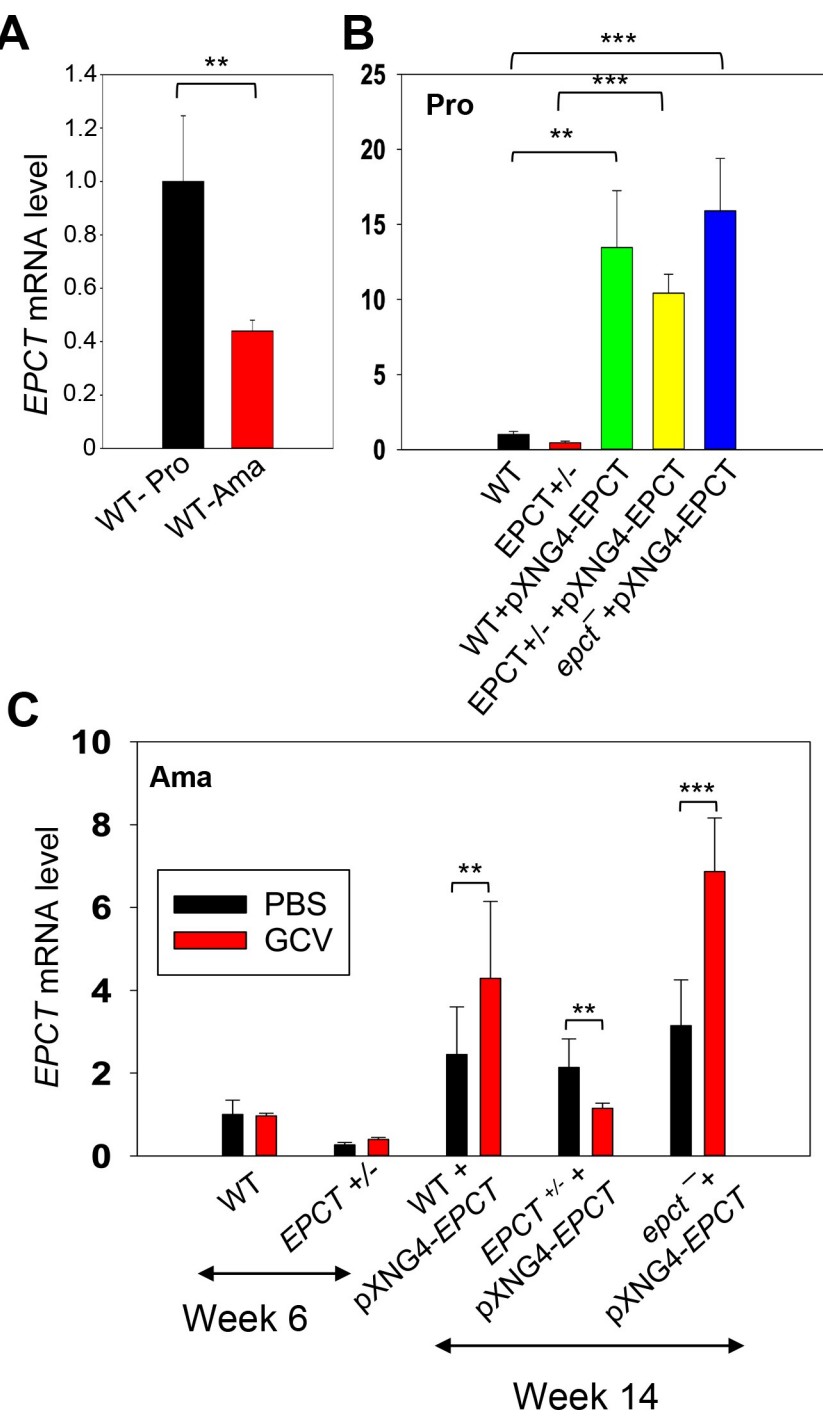

**Fig 6. Expression level of *EPCT* mRNA in promastigotes and amastigotes.** Total RNA was extracted from promastigotes (Pro) or amastigotes and the relative *EPCT* transcript levels were determined by qRT-PCR using the ΔΔCt method with the α-tubulin gene as the internal control. (**A**) WT promastigotes and WT amastigotes (6 weeks post infection). (**B**) Promastigotes grown in the absence (WT and *EPCT*+/-) or presence of SAT (WT+pXNG4-EPCT, EPCT+/- +pXNG4-EPCT and *epct*+pXNG4-EPCT). (**C**) Lesion derived amastigotes from mice treated with GCV or PBS (weeks 6 for WT and *EPCT*+/-, week 14–20 for EPCT overexpressors). Error bars represent standard deviation from three independent repeats (**: $p < 0.01$, ***: $p < 0.001$).

transcription from the endogenous *EPCT* locus when the plasmid was depleted (Figs 5C and 6C).

## Changes in EPCT expression affects the syntheses of glycerophospholipids and GP63

To examine if *EPCT* expression level affects phospholipid composition in *L. major*, total lipids were extracted from stationary phase promastigotes and analyzed by electrospray ionization mass spectrometry (ESI-MS). To detect PME and diacyl-PE, we used precursor ion scan of m/z 196 in the negative ion mode and added 14:0/14:0-PE as a standard for quantitation. As summarized in Fig 7A (details in S9 Fig), EPCT-overexpressing cells (WT +pXNG4-EPCT, EPCT +/- +pXNG4-EPCT and *epct*+ pXNG4-EPCT) had 30–50% less PME (*p*18:0/18:2- and *p*18:0/18:1-PE) relative to WT promastigotes. Meanwhile, the overall PME level in EPCT+/- was only slightly less than WT (due to reduced level of *p*18:0/18:1-PE), and there was no significant change in the abundance of diacyl-PE molecules (Figs 7A and S9). Similar ESI-MS analyses were performed to determine the levels of PC, phosphatidylinositol (PI), and IPC (the major sphingolipid in *Leishmania*). As summarized in Figs 7B and S10–S12, *EPCT* half knockout (EPCT+/-) or over-expression did not affect the composition of PC or IPC. Furthermore, *EPCT* overexpressing cells had 20–30% less 18:0/18:1-PI, the most abundant type of PI in comparison to WT cells (Figs 7C and S11). We also detected a ~70% reduction in the level of a18:0/18:1-PI (an alkyl-acyl PI) in WT+ pXNG4-*EPCT* parasites (Fig 7C). Collectively, these lipidomic analyses suggest that proper regulation of *EPCT* expression is vital for the balanced synthesis of glycerophospholipids.

Next, because PE and PI are involved in the biosynthesis of GP63 (a zinc-dependent, GPI-anchored metalloprotease) and lipophosphoglycan (LPG), respectively, we examined whether *EPCT* under- or over-expression could influence the production of these surface glycoconjugates. Interestingly, immunofluorescence microscopy revealed a 4-5-fold reduction of GP63 in EPCT+/-, WT +pXNG4-EPCT, and EPCT+/- +pXNG4-EPCT parasites and a nearly 8-fold reduction in *epct* + pXNG4-EPCT (Fig 8A and 8C). On the other hand, the cellular levels of LPG (determined by western blot) were unaltered in *EPCT* mutant lines (Fig 8B and 8D). These findings argue that *ECPT* expression from its chromosomal locus is required for the proper synthesis of GP63.

## EPCT overexpression affects stress response

To further explore how EPCT overexpression may compromise *L. major* virulence, we examined the response of mutants to starvation, acidic pH and heat stress as tolerance to these stress conditions are essential for parasite survival in the macrophage phagolysosome. Under normal conditions (complete M199 medium, pH7.4, 27˚C), EPCT+/- and overexpressors proliferated at similar rates as WT promastigotes in log phase and exhibited good viability in stationary phase (Figs 3E and 9A). However, if cells were transferred to phosphate-buffered saline (PBS, pH7.4) to test starvation tolerance, 50–60% of *EPCT* overexpressing cells (WT+pXNG4-EPCT, EPCT+/- +pXNG4-EPCT and *epct*+ pXNG4-EPCT) died after 2–3 days in comparison to <20% death for WT parasites (Fig 9B). These overexpressors also showed a 24–48-hour growth delay if they were cultivated in a pH5.0 medium (Fig 9C). In addition, *epct*+ pXNG4-*EPCT* parasites were slightly more sensitive to acidic stress (pH 5.0, Fig 9D) and heat (37˚C, Fig 9E and 9F) than WT in the stationary phase We did not detect any significant difference in mitochondrial superoxide level between WT and EPCT mutants (S13 Fig). Whether these defects are linked to the altered lipid contents in EPCT overexpressors remains to be determined. Regardless, sensitivity to stress conditions likely contributes to their lack of virulence.

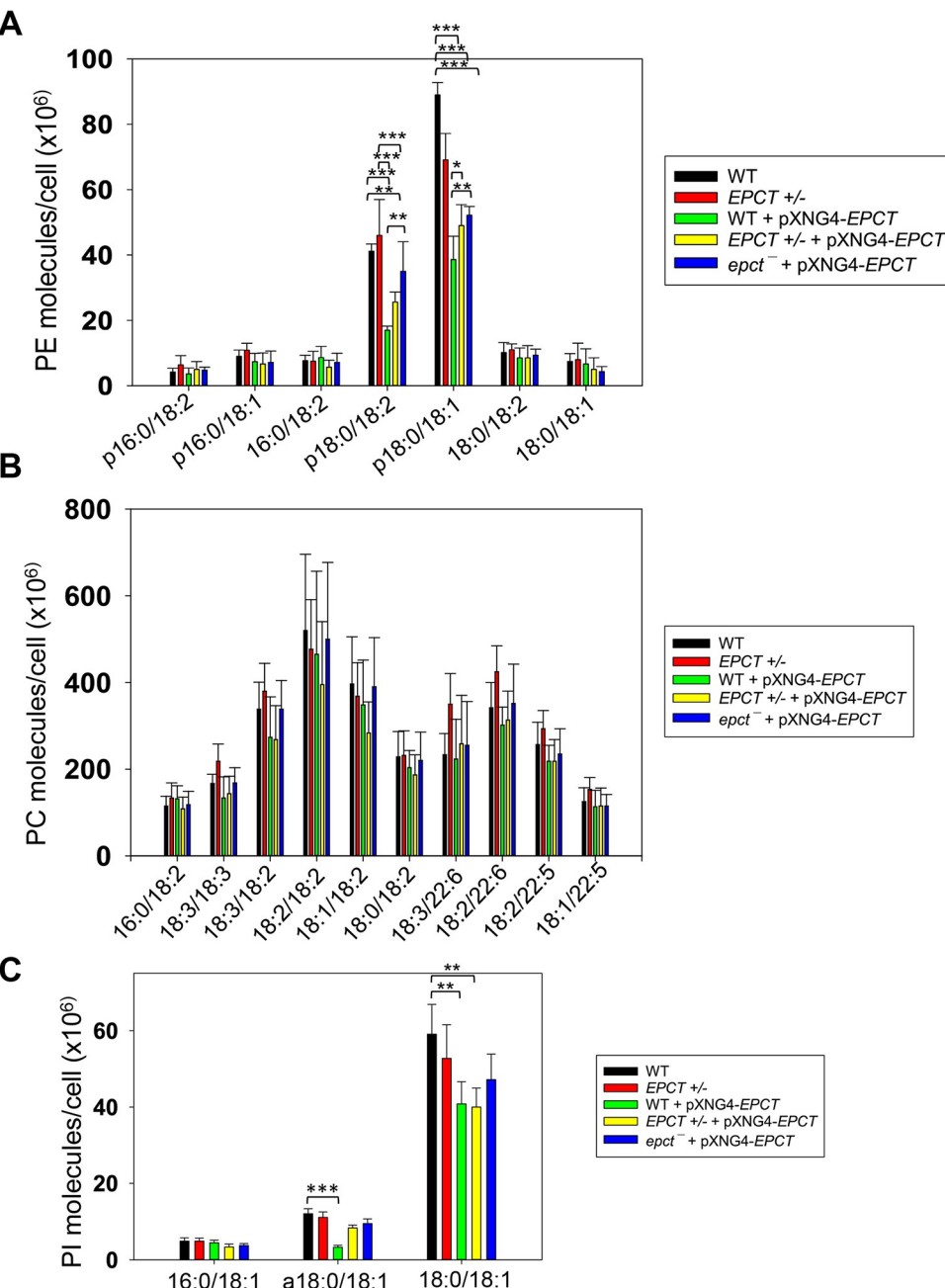

**Fig 7. EPCT overexpression alters lipid composition.** Total lipids were extracted from promastigotes and analyzed by ESI-MS as described in *Materials and Methods*. Cellular levels of PE (including PME and diacyl PE), PC, and PI were determined through comparison to internal standards and summarized in **A**, **B** and **C**, respectively. Error bars represent standard deviation from 5 independent experiments (*: $p < 0.05$, **: $p < 0.01$, ***: $p < 0.001$). More details were included in S9–S11 Figs).

## Discussion

Using a complementing episome assisted knockout approach coupled with negative selection, we demonstrate that EPCT is indispensable throughout the life cycle in *L. major*. The same method was applied to verify the essentiality of several genes in *Leishmania* [46–48]. The fact

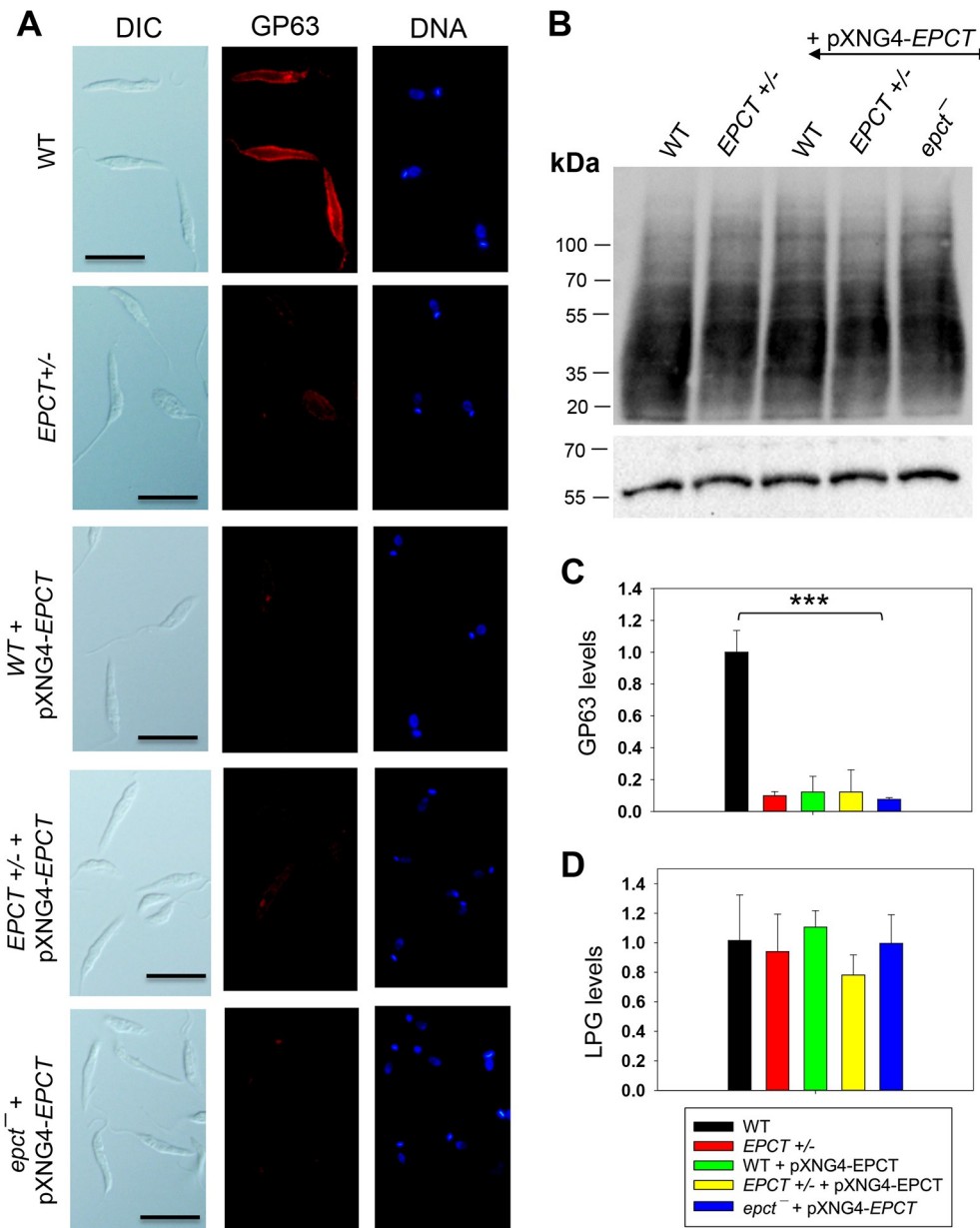

**Fig 8. EPCT overexpression leads to reduced level of GP63 but not LPG.** (**A**) WT, *EPCT*+/-, WT + pXNG4-*EPCT*, *EPCT*+/- + pXNG4-*EPCT*, and *epct⁻* + pXNG4-*EPCT* promastigotes were labeled with an anti-GP63 monoclonal antibody followed by an anti-mouse IgG-Texas Red antibody and DNA staining with Hoechst 33342. Scale bar: 10 μm. (**B**) Whole cell lysates from stationary phase promastigotes were analyzed by western blot using an anti-*L. major* LPG antibody (top) or anti-α-tubulin antibody (bottom). The relative abundances of GP63 (**C**) and LPG (**D**) were determined and normalized to WT levels as described in *Materials and Methods*. Error bars represent standard deviations from 3 independent repeats (***: $p < 0.001$).

that EPCT is essential in both promastigotes and amastigotes is in sharp contrast to choline-phosphate cytidylyltransferase (CPCT) which catalyzes the equivalent step in the choline branch of the Kennedy pathway by combining choline phosphate and CTP into CDP-choline (Fig 1). *L. major* CPCT-null mutants (*cpct⁻*) cannot incorporate choline into PC but contain similar levels of PC as WT parasites [49]. Loss of CPCT has no impact on promastigote growth

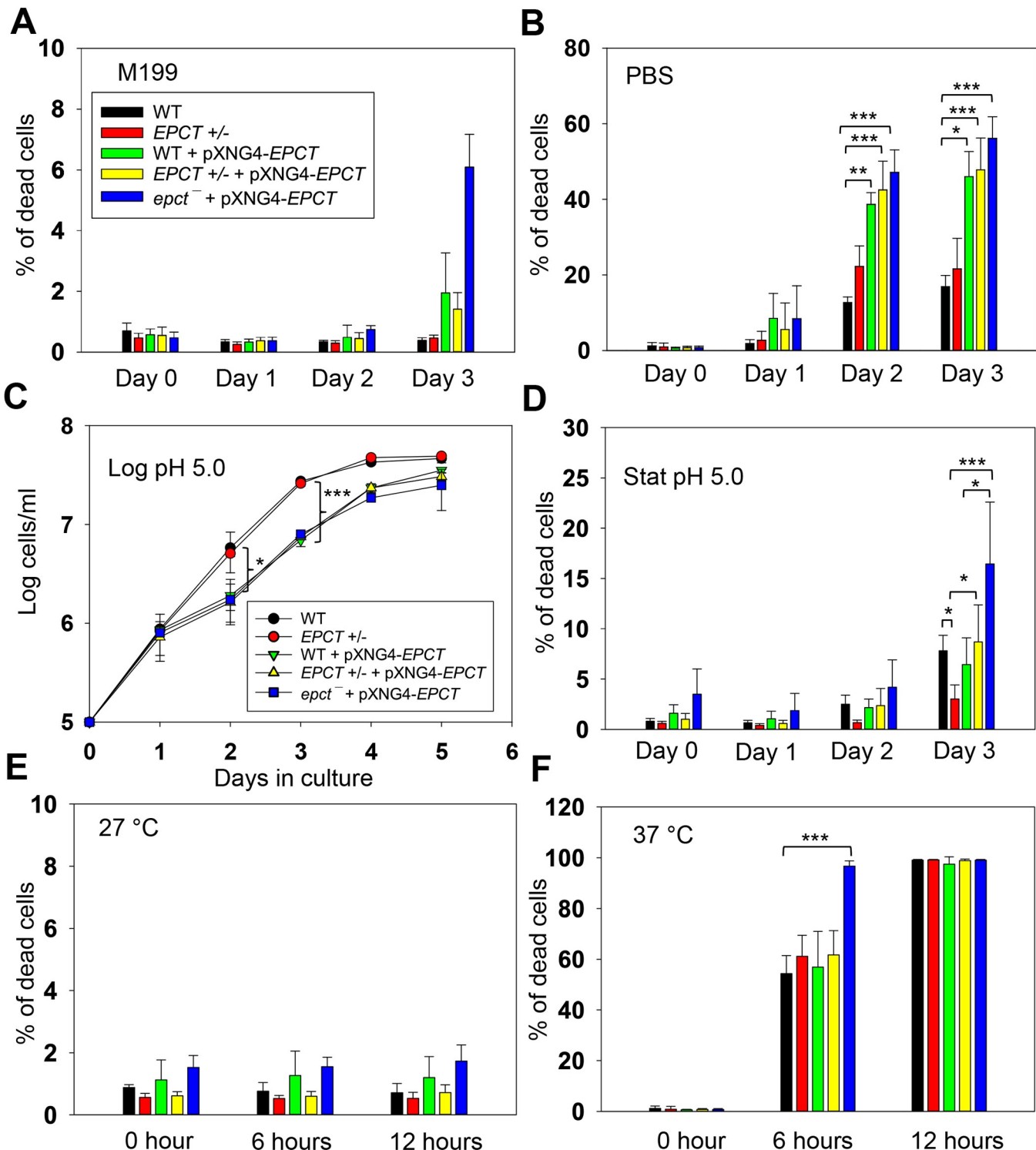

**Fig 9. EPCT overexpression leads to heightened sensitivity to acidic pH and starvation conditions.** (**A-B**) Promastigotes were cultivated in regular M199 media (pH 7.4) at 27˚C until reaching stationary phase. Cells were then kept in M199 media (**A**) or transferred into phosphate-buffered saline (PBS) and cell viability was examined using flow cytometry ((**B**). (**C-D**) Promastigotes were cultivated in an acidic M199 medium (pH5.0). Cell growth over time was determined by hemocytometer counting (**C**) and cell viability during stationary phase was monitored by flow cytometry (**D**). (**E-F**) Day 1 stationary phase promastigotes cultivated in regular M199 media (pH 7.4) were incubated at 27˚C (**E**) or 37˚C (**F**). Percentages of dead cells were determined by flow cytometry after propidium iodide staining (**A**, **B**, **D**-**F**). Error bars represent standard deviations from 3–4 independent experiments (*: $p < 0.05$, **: $p < 0.01$, ***: $p < 0.001$).

in culture or their virulence in mice [49]. A similar study on CEPT which is directly responsible for the generation of diacyl-PE and PC revealed that this enzyme is only required for the promastigote but not amastigote stage [18]. These findings indicate that intracellular amastigotes can survive and proliferate without *de novo* PC synthesis by relying on the uptake/remodeling of host lipids.

Like with CEPT, chromosomal null mutants for *EPCT* cannot lose the complementing pXNG4-EPCT episome in culture, suggesting that *de novo* PE synthesis is required for the fast-replicating promastigotes (Fig 4). When the GFP-low cells were separated from GCV-treated *epct*+ pXNG4-EPCT promastigotes by FACS, they quickly regained GFP expression, suggesting that these cells can only survive and proliferate in the presence of GFP-high cells (S5 Fig). *Leishmania* parasites are known to produce and exchange extracellular vesicles among individual cells [50] which may contribute to the survival of *epct*+ pXNG4-EPCT parasites after GCV treatment.

The fact that EPCT is indispensable for amastigotes is intriguing, as amastigotes can acquire enough PC, which is far more abundant than PE [18]. One possibility is that the PE scavenging pathway, either direct uptake or remodeling of host lipids, is insufficient and amastigotes need certain level of *de novo* synthesis to meet the demand for PE. Alternatively, because of its central position in the Kennedy pathway (Fig 1), a loss of EPCT will not only abolish the *de novo* synthesis of PME and diacyl-PE, but also negatively affect the production of PC (from PE N-methylation) and PS (from PE interconversion). In comparison to EPT, CPCT and CEPT, disruption of EPCT would result in a more pleiotropic effect on the synthesis of multiple classes of glycerophospholipids (Fig 1).

We observed a significant virulence attenuation from EPCT overexpression as parasites with pXNG4-ECPT or pGEM-EPCT showed delayed lesion progression and cell replication in mice (Figs 5 and S8). Such defects were not found with the episomal overexpression EPT, CPCT, CEPT, or FPPS (an essential enzyme required for sterol synthesis) [14,18,48,49]. In mammalian cells, endogenous EPCT is reported to have a bimodal distribution between the cytosol and smooth endoplasmic reticulum (ER) [51]. EPCT generates CDP-EtN for EPT and CEPT to synthesize PME and diacyl-PE, respectively (Figs 1 and 2). Both EPT and CEPT are localized in the ER [14,18]. Our results indicate that EPCT overexpression caused reduction in PME but had little effect on diacyl-PE or PC (Figs 7, S9 and S10). Perhaps EPCT overexpression (mainly in the cytosol) led to increased production of CDP-EtN in cytosol but not ER which reduced PE synthesis. Alternatively, the elevated level of CDP-EtN may cause substrate inhibition when substrate concentrations exceed the optimum level for certain enzymes which hinders product release [52]. We did not detect any obvious alteration in cell volume/shape or compensatory changes in the lipid composition of EPCT over-expressors based on our lipidomic analysis of PE, PC, PI and IPC. Effects of EPCT overexpression on neutral lipids (e.g., sterols, DAG and TAG) remain to be determined.

It is not clear whether the altered PME synthesis in EPCT overexpressing cells is responsible for their heightened sensitivity to starvation or acidic pH, although these defects likely contribute to the virulence attenuation in mice. Our previous report on *L. major ept*- mutants indicate that while these mutants are largely devoid of PME, they did not exhibit the same stress response defects as EPCT overexpressing cells [14]. It is possible that the increased cytosolic concentration of CDP-EtN, coupled with the depletion of EtN-P and CTP, causes cytotoxicity when parasite encounter starvation or acidic stress. Another possibility is that reduced PME synthesis compromised macroautophagy (a process required the conjugation of PE to ATG8 [9]) in EPCT overexpressing cells, resulting in hypersensitivity to starvation. We did not detect significant changes in mitochondrial superoxide production from EPCT under- or over-expression, suggesting that mitochondrial PE synthesis is not affected (S13 Fig).

Consistent with the detrimental effect of EPCT overexpression on amastigote growth, we found a steady decline of pXNG4-EPCT copy number in WT +pXNG4-EPCT and *EPCT*+/- +-pXNG4-EPCT amastigotes (Fig 5C). GCV treatment of mice caused further reduction in plasmid level as expected (Fig 5C). Curiously, neither *EPCT*+/- +pXNG4-EPCT amastigotes nor WT +pXNG4-EPCT amastigotes could completely lose the episome like *EPCT*+/- +-pXNG4-EPCT promastigotes in culture after GCV treatment (Figs 4A and 5C). Our previous studies on farnesyl pyrophosphate synthase (FPPS) and CEPT have demonstrated that heterozygous mutants of those genes lost most of their respective episome (<0.2 copy per cell for *FPPS*+/− +pXNG4-FPPS and *CEPT*+/− +pXNG4-CEPT) within 6 weeks post infection [18,48]. It is possible that the slow replication rates of *EPCT*+/- +pXNG4-EPCT and WT +-pXNG4-EPCT amastigotes prevented complete plasmid loss.

For *epct*+ pXNG4-EPCT amastigotes, they retained 12–24 plasmids per cell throughout the course of infection, despite the cytotoxic effects from GCV and EPCT overexpression (Fig 5C). We also detected elevated *EPCT* transcript from *epct*+ pXNG4-EPCT amastigotes isolated from GCV-treated mice (Fig 6C). These findings suggest that the need for EPCT outweighs the negative impacts from GCV and EPCT overexpression for *L. major* during the mammalian stage. However, it is important to remember that GCV treatment of mice only lasted for two weeks, and the level of GCV might be below the minimum effective concentration in the later stage of infection. In addition, plasmid copy number and EPCT transcript level may not fully reflect the EPCT protein level.

While the alteration of *EPCT* expression had little impact on the cellular level of lipophosphoglycan (LPG), it did significantly reduce the expression of GP63, a zinc-dependent metalloprotease that plays pivotal role *Leishmania* infection through proteolytic cleavage of host complement proteins and subversion of macrophage signaling [53–56]. Synthesis of GP63 requires PE to serve as a donor to generate the EtN-P linkage between protein and GPI-anchor [7,57]. It is not known whether a specific subset of PE (PME or di-acyl PE) is preferentially used in GP63 synthesis. As illustrated in Fig 8A and 8C, *EPCT*+/- parasites contained only 10–15% of WT-level GP63, and the episomal expression of *EPCT* did not rescue this defect. These results suggest that *EPCT* must be expressed from its endogenous locus to fully support GP63 synthesis, raising the possibility that endogenously expressed EPCT is modified or compartmentalized differently from episomally expressed EPCT, as previously reported for other leishmanial proteins [58].

In summary, our study establishes EPCT as the most important enzyme in the Kennedy pathway. It is crucial for *L. major* survival during the promastigote and amastigote stages. Overexpression of EPCT alters lipid homeostasis and stress response, leading to severely attenuated virulence. Future studies will investigate how intracellular amastigotes balance *de novo* synthesis with scavenging to optimize their long-term survival and evaluate the potential of EPCT as a new anti-*Leishmania* target.

## Materials and methods

### Ethics statement

All procedures involving live mice were performed as per approved protocol by the Animal Care and Use Committee at Texas Tech University (PHS Approved Animal Welfare Assurance No A3629-01).

### Materials

Lipid standards for mass spectrometry studies including 1,2-dimyristoyl-sn-glycero-3-phosphoethanolamine (14:0/14:0-PE), 1,2-dimyristoyl-sn-glycero-3-phosphocholine (14:0/

14:0-PC), and 1,2-dipalmitoyl-sn-glycero-3-phosphoinositol (16:0/16:0-PI) were purchased from Avanti Polar Lipids (Alabaster, AL). For the EPCT activity assay, phosphoryl ethanol-amine [1,2-$^{14}$C] (50–60 mCi/mmol) or [$^{14}$-C] EtN-P was purchased from American Radiola-beled Chemicals (St Louis, MO). All other reagents were purchased from ThermoFisher Scientific or Sigma Aldrich Inc unless otherwise specified.

### *Leishmania* culture and genetic manipulations

*Leishmania major* strain FV1 (MHOM/IL/81/Friedlin) promastigotes were cultivated at 27˚C in a complete M199 medium (M199 with 10% heat inactivated fetal bovine serum and other supplements, pH 7.4) [59]. To monitor growth, culture densities were measured daily by hemocytometer counting. Log phase promastigotes refer to replicative parasites at densities <1.0 x 10$^7$ cells/ml, and stationary phase promastigotes refer to non-replicative parasites >2.0 x 10$^7$ cells/ml.

To delete chromosomal *EPCT* alleles, the upstream and downstream flanking sequences (~1 Kb each) of *EPCT* were amplified by PCR and cloned in the pUC18 vector. Genes confer-ring resistance to blasticidin (*BSD*) and puromycin (*PAC*) were then cloned between the upstream and downstream flanking sequences to generate pUC18-KO-EPCT:BSD and pUC18-KO-EPCT:PAC, respectively. To generate the *EPCT+/-* heterozygotes (*ΔEPCT::BSD/ EPCT*), wild type (WT) *L. major* promastigotes were transfected with linearized *BSD* knockout fragment (derived from pUC18-KO-EPCT:BSD) by electroporation and transfectants showing resistance to blasticidin were selected and later confirmed to be *EPCT+/-* by Southern blot as previously described [18]. To delete the second chromosomal allele of *EPCT*, we used an epi-some assisted approach as previously described [46]. First, the *EPCT* open reading frame (ORF) was cloned into the pXNG4 vector to generate pXNG4-EPCT and the resulting plasmid was introduced into *EPCT+/-* parasites. The resulting *EPCT+/-* +pXNG4-EPCT cell lines were then transfected with linearized *PAC* knockout fragment (derived from pUC18-KO-EPCT: PAC) and selected with 15 μg/ml of blasticidin, 15 μg/ml of puromycin and 150 μg/ml of nour-seothricin. The resulting *EPCT* chromosomal null mutants with pXNG4-EPCT (*ΔEPCT::BSD/ ΔEPCT::PAC* + pXNG4-EPCT or *epct*+ pXNG4-EPCT) were validated by Southern blot as described in Figs 3, S3 and S4. For EPCT activity assay and localization studies, the *EPCT* ORF was into the pXG vector or pXG-GFP' vector to generate pXG-EPCT or pXG-GFP-EPCT respectively, followed by their introduction into WT *L. major* promastigotes by electropora-tion. Finally, a pGEM-5'-Phleo-DST IR-EPCT-3' construct was generated by cloning the *EPCT* ORF along with its upstream- and downstream flanking sequences (~1 Kb each) into a pGEM vector [60]. The resulting plasmid was introduced into *EPCT+/-* to drive *EPCT* overex-pression in the presence of its flanking sequences.

### EPCT activity assay

EPCT assay was adopted based on a previously protocol [35]. Log phase promastigotes of WT, WT + pXG-EPCT (WT+EPCT), and WT + pXG-GFP-EPCT (WT+GFP-EPCT) were resus-pended in a lysis buffer based phosphate-buffered saline (PBS, pH 7.4) with 5 mM MgCl2, 0.1% Triton X-100, 5 mM DTT, and 1 X protease inhibitor cocktail at 4 x 10$^8$ cells/ml. *Leish-mania* lysate was incubated with 1 μCi of [$^{14}$-C] EtN-P in a reaction buffer (100 mM Tris pH 7.5, 10 mM MgCl2, and 5 mM CTP) at room temperature for 20 min, boiled at 100˚C for 5 min to stop the reaction, and loaded directly on a Silica gel 60 TLC plate (10 μl each). TLC was developed in 100% ethanol:0.5% NaCl:25% ammonium hydroxide (10:10:1, v/v) and signals were detected via autoradiography at -80˚C. Ratios of [$^{14}$-C] CDP-EtN]/[$^{14}$-C] EtN-P were determined using Adobe Photoshop. Mouse liver homogenate (with similar protein

concentration as *Leishmania* lysate) was used as a positive control whereas boiled *Leishmania* lysate and mouse liver homogenate were included as negative controls. To determine the mobility of EPCT substrate (EtN-P) and product (CDP-EtN), non-radiolabeled EtN-P and CDP-EtN (40 nmol each) were loaded onto a Silica gel 60 TLC plate and processed with the same solvent as described above, followed by 0.2% ninhydrin spray.

## Sub-cellular fractionation

Promastigotes were fractionated following a protocol adapted from *T. cruzi* work [61]. Briefly, mid-log cells were harvested by centrifugation and washed in PBS. Cells were resuspended in a lysis buffer (20 mM Tris-HCl pH 7.5, 150 mM NaCl, 0.1% v/v Tween-20, 2 x protease inhibitor) at 2 x $10^8$ cells/ml. After 15 min incubation on ice (vortex every 2 min), cell lysates were cleared (1000g, 10 min, 4˚C) and subjected to sequential centrifugation at 10,000g (10 min, 4˚C) and then 100,000g (60 min, 4˚C) to isolate the cytosolic fraction. Pellets from 10,000g centrifugation (large granules) and 100,000g centrifugation (microsomal fraction) were resuspended in lysis buffer to the same volume as that of the cytosolic fraction. Western blot was performed as described below.

## Fluorescence microscopy and western blot

An Olympus BX51 Upright Epifluorescence Microscope equipped with a digital camera was used to visualize the localization of GFP-EPCT and GP63 as previously described [14]. For GFP-EPCT, WT + pXG-GFP-EPCT promastigotes were attached to poly-L-lysine coated coverslips and fixed with 3.7% paraformaldehyde prior to visualization. GP63 staining of unpermeabilized parasites was performed using an anti-GP63 monoclonal antibody (#96/126, 1:500, Abcam Inc.) at ambient temperature for 30 min, followed by washing and incubation with a goat-anti-mouse-IgG-Texas Red antibody (1:500) for 30 min. DNA staining was performed using Hoechst 33342 (2.5 µg/ml) for 10 min. All images were acquired using the same exposure times (GFP: 60 ms, DNA: 40 ms, DIC: 20 ms) and processed with the same settings in Adobe Photoshop. Because the anti-GP63 monoclonal antibody (#96/126) does not recognized denatured GP63 on western blot, we determined the cellular level of GP63 by dividing the integrated density of GP63 fluorescence (minus background) with the integrated density of Hoechst (minus background). For each cell type, 200 cells were analyzed, and the average of GP63/Hoechst was determined and normalized to WT (as 1.0).

To examine the cellular level of lipophosphoglycan (LPG) and GFP-EPCT, promastigote lysates were heated in SDS sample buffer at 65˚C for 5 min and resolved by SDS-PAGE. After transfer to PVDF membrane, blots were probed with mouse anti-LPG monoclonal antibody WIC79.3 (1:1000) [62] followed by goat anti-mouse IgG conjugated with HRP (1:2000). GFP-EPCT was detected using a rabbit anti-GFP antibody (Abcam Inc., 1:2000) followed by goat anti-rabbit IgG-HRP (1:2000). Antibody to alpha-tubulin was used as the loading control. For sub-cellular fractions, 6 µg of cytosolic fraction, large granules, and microsomal fraction were probed with anti-GFP, -LPG or -HSP83 antibodies.

## Promastigote essentiality assay

*EPCT*+/- +pXNG4-EPCT and *epct̄*+ pXNG4-EPCT promastigotes were inoculated in complete M199 media at 1.0 x $10^5$ cells/ml in the presence or absence of 50 µg/ml of GCV (the negative selection agent) or 150 µg/ml of nourseothricin (the positive selection agent). Every three days, cells were reinoculated into fresh media with the same negative or positive selection agents and percentages of GFP-high cells for each passage were determined by flow cytometry using an Attune NxT Acoustic Flow Cytometer. After 14 passages, GFP-low cells were isolated

from *EPCT+/-* +pXNG4-EPCT and *epct̄*+ pXNG4-EPCT by fluorescence-activated cell sorting (FACS). Individual clones were then isolated via serial dilution in 96-well plates and expanded in the presence of GCV and absence of nourseothricin. The GFP levels and pXNG4-EPCT plasmid copy numbers of selected clones were determined by flow cytometry and qPCR, respectively.

## Mouse footpad infection and *in vivo* GCV treatment

Female BALB/c mice (7–8 weeks old) were purchased from Charles River Laboratories International (Wilmington, MA). To determine whether EPCT is required during the intracellular amastigote stage, day 3 stationary phase promastigotes were injected into the left hind footpad of BALB/c mice ($1.0 \times 10^6$ cells/mouse, 10 mice per group). For each group, starting from day one post infection, one half of the mice received GCV at 7.5 mg/kg/day for 14 consecutive days (0.5 ml each, intraperitoneal injection), while the other half (control group) received equivalent volume of sterile PBS. Footpad lesions were measured weekly using a Vernier caliper after anesthetization with isoflurane (air flow rate: 0.3–0.5 ml/hour). Mice were euthanized through controlled flow of $CO_2$ asphyxiation when lesions reached 2.5 mm (humane endpoint) or upon the onset of secondary infections. Parasite numbers in infected footpads were determined by limiting dilution assay [63] and qPCR as described below.

## Quantitative PCR (qPCR) analysis

To determine parasite loads in infected footpads, genomic DNA was extracted from footpad homogenate and qPCR reactions were run in triplicates using primers targeting the 28S rRNA gene of *L. major* [18]. Cycle threshold (Ct) values were determined from melt curve analysis. A standard curve of Ct values was generated using serially diluted genomic DNA samples from *L. major* promastigotes (from 0.1 cell/reaction to $10^5$ cells/reaction) and Ct values >30 were considered negative. Parasite numbers in footpad samples were calculated from their Ct values using the standard curve. Control reactions included sterile water and DNA extracted from uninfected mouse liver.

To determine pXNG4-*EPCT* plasmid levels in promastigotes and lesion-derived amastigotes, a similar standard curve was generated using serially diluting pXNG4-EPCT plasmid DNA (from 0.1 copy/reaction to $10^5$ copies/reaction) and primers targeting the *GFP* region. qPCR was performed with the same set of primers on DNA samples and the average plasmid copy number per cell was determined by dividing total plasmid copy number with total parasite number based on Ct values.

To determine EPCT transcript levels, total RNA was extracted from promastigotes or lesion-derived amastigotes and converted into cDNA using a high-capacity reverse transcription kit (Bio-Rad), followed by qPCR using primers targeting *EPCT* or α-tubulin coding sequences. The relative expression level of *EPCT* was normalized to that of α-tubulin using the $2^{-\Delta\Delta(Ct)}$ method [64]. Control reactions were carried out without leishmanial RNA and without reverse transcriptase.

## Lipid extraction and mass spectrometric analyses

Total lipids from stationary phase promastigotes ($1.0 \times 10^8$ cells/sample) were extracted using the Bligh & Dyer method [65]. Commercial non-indigenous lipid standards were added to cell lysates as internal standards at the time of lipid extraction ($1.0 \times 10^8$ molecules/cell for 14:0/14:0-PE, $5.0 \times 10^7$ molecules/cell for 14:0/14:0-PC, and $1.0 \times 10^8$ molecules/cell for 16:0/16:0-PI). These lipid standards are absent from *L. major* promastigotes. Determination of lipid families by electrospray ionization mass spectrometry (ESI-MS) was carried out by a Thermo

Vantage TSQ instrument applying precursor ion scan of m/z 196 for PE, precursor ion scan of m/z 241 for PI and IPC in the negative-ion mode, and precursor ion scan of m/z 184 for PC in the positive-ion mode [66]. Individual lipid species and their structures were also confirmed by high resolution mass spectrometry performed on a Thermo LTQ Obitrap Velos with a resolution of 100,000 (at m/z 400). All lipidomic analyses were performed five times.

### *Leishmania* stress response assays and determination of mitochondrial ROS level

For heat tolerance, *L. major* promastigotes grown in complete M199 media were incubated at either 27˚C or 37˚C. To test their sensitivity to acidic pH, promastigotes were transferred to a pH5.0 medium (same as the complete M199 medium except that the pH was adjusted to 5.0 using hydrochloric acid). For starvation response, promastigotes were transferred to PBS (pH 7.4). Cell viability was determined at the indicated times by flow cytometry after staining with 5 μg/ml of propidium iodide. Parasite growth was monitored by cell counting using a hemocytometer.

Mitochondrial superoxide accumulation was determined as described previously [34]. Briefly, log phase promastigotes were transferred to PBS (pH 7.4) and stained with 5 μM of MitoSox Red. After 25 min incubation at 27˚C, the mean fluorescence intensity (MFI) for each sample was measured by flow cytometry.

### Statistical analysis

Unless otherwise specified, experiments were repeated three to five times. Symbols or bars represent averaged values and error bars represent standard deviations. Differences between groups were assessed by one-way ANOVA (for three or more groups) using the Sigmaplot 13.0 software (Systat Software Inc, San Jose, CA) or student $t$ test (for two groups). $P$ values were grouped in all figures (\*\*\*: $p < 0.001$, \*\*: $p < 0.01$, \*: $p < 0.05$).

### Supporting information

**S1 Fig. Alignment of EPCT amino acid sequences from *Saccharomyces cerevisiae* (*Sc*, Genbank: P33412), *Plasmodium falciparum* (*Pf*, Plasmodb: PfDd2_130053500), *Leishmania major* (*Lm*, Tritrypdb: LmjF.32.0890), *Arabidopsis thaliana* (*At*, TAIR: AT2G38670.1), and *Homo sapiens* (*Hs*, Genbank: Q99447) using Clustal Omega.** Asterisks (\*): fully conserved residues; colons (:) highly similar residues; periods (.): moderately similar residues. Color code for amino acids: red-nonpolar; green-polar; blue-acidic; purple-basic. (PDF)

**S2 Fig. Detection of EPCT activity with thin layer chromatography (TLC).** (**A**) EtN-P or CDP-EtN (40 nmol each) are resolved by TLC as described in *Materials and Methods*. The plate was dried and sprayed with 0.2% ninhydrin to show the positions of EtN-P and CDP-EtN. (**B**) Mouse liver lysates (boiled and not boiled, two repeats each) were incubated with [$^{14}$C]-EtN-P at room temperature, followed by TLC analysis and signals were detected by autoradiography. (PDF)

**S3 Fig. Chromosomal *EPCT*-null mutants cannot be generated without a complementing episome.** Genomic DNA samples from *L. major* FV1 WT, LV39 WT, *EPCT*+/- (#1 and #2), and putative *epct*⁻ (#1D, #1E, and #1F) parasites were digested by *Hind* III + *Bam* HI followed by Southern blot analyses using radiolabeled probes for an upstream flanking sequence (5' probe: **A** and **C**) and the open reading frame of *EPCT* (ORF probe: **B** and **D**). The approximate

recognition sites of *Hind* III and *Bam* HI and expected DNA fragment sizes are indicated in **E**.
(PDF)

**S4 Fig. Scheme of the Southern blot in Fig 3.** Expected DNA fragment sizes for using the 5'
probe (**A**-**B**) or ORF probe of *EPCT* (**C**-**D**) are indicated. TK: thymidine kinase, GFP: green
fluorescent protein, SAT: nourseothricin resistance gene.
(PDF)

**S5 Fig. Attempts to isolate GFP-low cells from *epct̄*+pXNG4-EPCT by FACS.** (**A**) *Epct̄*+-
pXNG4-EPCT promastigotes were cultivated in the presence of GCV and absence of SAT for
14 passages as indicated in Fig 4D. GFP-low population (**B**) was separated from GFP-high
population (**C**) by FACS. Single clones were then isolated from the GFP-low population (**B**)
via serial dilution followed by two consecutive rounds of expansion in the presence of GCV
and absence of SAT (**D** and **E** represent the first and second round of expansion respectively).
GFP expression levels in **A**-**E** were analyzed by flow cytometry.
(PDF)

**S6 Fig. Chromosomal *EPCT*-null promastigotes cannot lose the pXNG4-EPCT episome.**
(**A**) *EPCT*+/- + pXNG4-EPCT and *epct̄* + pXNG4-EPCT promastigotes were cultivated in the
presence of SAT or GCV for 14 passages (as pools) and individual clones were isolated via
FACS followed by serial dilution as described in *Materials and Methods*. Plasmid copy number
numbers (average ± SDs) were determined by qPCR. (**B**) Promastigotes were cultivated at
27˚C in complete M199 media and culture densities were determined daily using a hemocy-
tometer. Error bars indicate standard deviations from three biological repeats (*: $p < 0.05$, **:
$p < 0.01$).
(PDF)

**S7 Fig. EPCT overexpression leads to reduced growth in BALB/c mice.** Following footpad
infection, mice were treated with GCV or PBS and euthanized at the indicated timepoints. (**A**)
Mouse body weights were measured at day 0–21 post infection. (**B**) Genomic DNA samples
were prepared from lesion-derived amastigotes and parasite loads were determined by qPCR
using primers targeting the *L. major* 28S rDNA gene (*: $p < 0.05$, **: $p < 0.01$, ***: $p < 0.001$).
(PDF)

**S8 Fig. EPCT overexpression leads to attenuated virulence in BALB/c mice.** Stationary
phase promastigotes were injected into the footpad of BALB/c mice as described in *Materials
and Methods*. Footpad lesion sizes were measured using a Vernier caliper (**A**) and parasite
loads were determined by qPCR (**B**). **: $p < 0.01$.
(PDF)

**S9 Fig. EPCT overexpression leads to reduced levels of PME.** Total lipids were extracted
from log phase promastigotes and analyzed by ESI-MS in the negative ion mode as described
in *Materials and Methods*. The 14:0/14:0-PE (m/z 634.5) was added as an internal standard.
Representative tandem mass spectra obtained from precursor ion scan of m/z 196 specifically
monitoring PME and diacyl-PE species were shown for WT, *EPCT*+/-, WT + pXNG4-*EPCT*,
*EPCT*+/- + pXNG4-*EPCT*, and *epct̄* + pXNG4-*EPCT*. Major PME species such as *p*18:0/
18:2-PE (m/z 726.5) and *p*18:0/18:1-PE (m/z 728.5) were indicated.
(PDF)

**S10 Fig. EPCT overexpression does not affect the synthesis of PC.** Total lipids were
extracted from log phase promastigotes and analyzed by ESI-MS in the positive ion mode as
described in *Materials and Methods*. The 14:0/14:0-PC (m/z 678.5) was added as an internal

standard. Representative tandem mass spectra obtained from precursor ion scan of m/z 184 specifically monitoring PC and sphingomyelin species were shown for WT, *EPCT+/-*, WT + pXNG4-*EPCT*, *EPCT+/-* + pXNG4-*EPCT*, and *epct̄* + pXNG4-*EPCT*. Major PC species such as 18:2/18:2-PC (m/z 782.6) and 18:2/22:6-PC (m/z 830.6) were illustrated.
(PDF)

**S11 Fig. EPCT overexpression leads to reduced levels of PI.** Total lipids were extracted from log phase promastigotes and analyzed by ESI-MS in the negative ion mode as described in *Materials and Methods*. The 16:0/16:0-PI (m/z: 809.6) was added as an internal standard. Representative tandem mass spectra obtained from precursor ion scan of m/z 241 (specific for PI) were shown for WT, *EPCT+/-*, WT + pXNG4-*EPCT*, *EPCT+/-* + pXNG4-*EPCT*, and *epct̄* + pXNG4-*EPCT*. Major PI species were indicated.
(PDF)

**S12 Fig. EPCT overexpression does not affect the cellular levels of IPC.** Total lipids were extracted from stationary phase promastigotes and analyzed by ESI/MS in the negative ion mode using both total ion current scan and precursor ion scan of m/z 241. Error bars represent standard deviations from 4 independent experiments.
(PDF)

**S13 Fig. EPCT overexpression does not affect mitochondrial ROS production.** Log phase promastigotes were cultivated in complete M199 medium **(A, C)** or transferred to PBS **(B, D)** and labeled with MitoSox Red for 25 min at ambient temperature. Mean fluorescence intensity (MFI) for MitoSox Red (**A**, **B**) and percentages of dead cells (**C**, **D**) were determined by flow cytometry at the indicated timepoints. Cell growth rates in M199 or PBS were determined by hemocytometer counting **(E)**. Error bars represent standard deviations from three independent experiments.
(PDF)

**S1 Table. List of oligonucleotides used in this study.** Sequences in lowercase represent restriction enzyme sites.
(PDF)

## Acknowledgments

Rabbit antibody against *Leishmania* HSP83 was a kind gift from Dr. Dan Zilberstein (Technion Israel Institute of Science).

## Author Contributions

**Conceptualization:** Somrita Basu, Mattie C. Pawlowic, Kai Zhang.

**Data curation:** Somrita Basu, Mattie C. Pawlowic, Fong-Fu Hsu.

**Formal analysis:** Somrita Basu, Fong-Fu Hsu, Kai Zhang.

**Funding acquisition:** Fong-Fu Hsu, Kai Zhang.

**Investigation:** Somrita Basu, Mattie C. Pawlowic, Fong-Fu Hsu, Geoff Thomas.

**Supervision:** Kai Zhang.

**Validation:** Somrita Basu, Kai Zhang.

**Writing – original draft:** Somrita Basu, Kai Zhang.

**Writing – review & editing:** Somrita Basu, Mattie C. Pawlowic, Fong-Fu Hsu, Geoff Thomas, Kai Zhang.

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
