## [Decision Letter · Decision Letter 0]

13 Mar 2023

Dear Dr. Zhang,

Thank you very much for submitting your manuscript "Ethanolaminephosphate cytidyltransferase is essential for survival, lipid homeostasis and stress tolerance in Leishmania major" for consideration at PLOS Pathogens. As with all papers reviewed by the journal, your manuscript was reviewed by members of the editorial board and by several independent reviewers. In light of the reviews (below this email), we would like to invite the resubmission of a significantly-revised version that takes into account the reviewers' comments.

The three reviewers that have evaluated your manuscript agree that an interesting pathway and enzyme has been addressed, and a significant number of experiments have been executed with many convincing outcomes. It is particularly appreciated that the results are in contrast to current thinking of lipid uptake and that EPCT might be an attractive drug target for leishmaniasis. However, a number of concerns are raised with respect to consistency of data, data interpretation and methodology as well as clarity of writing. These issues need to be carefully and comprehensively addressed. Many of the inconsistencies mentioned by the reviewers are apparently due to insufficient methodology or statistics and therefore need additional quantifications and orthogonal experiments. Also some of the reviewer´s questions can only be answered by additional experimental data as suggested.

We cannot make any decision about publication until we have seen the revised manuscript and your response to the reviewers' comments. Your revised manuscript is also likely to be sent to reviewers for further evaluation.

Sincerely,

Michael Boshart, M.D.

Academic Editor

PLOS Pathogens

Margaret Phillips

Section Editor

PLOS Pathogens

Kasturi Haldar

Editor-in-Chief

PLOS Pathogens

orcid.org/0000-0001-5065-158X

Michael Malim

Editor-in-Chief

PLOS Pathogens

orcid.org/0000-0002-7699-2064

The three reviewers that have evaluated your manuscript agree that an interesting pathway and enzyme has been addressed, and a significant number of experiments have been executed with many convincing outcomes. It is particularly appreciated that the results are in contrast to current thinking of lipid uptake and that EPCT might be an attractive drug target for leishmaniasis. However, a number of concerns are raised with respect to consistency of data, data interpretation and methodology as well as clarity of writing. These issues need to be carefully and comprehensively addressed. Many of the inconsistencies mentioned by the reviewers are apparently due to insufficient methodology or statistics and therefore need additional quantifications and orthogonal experiments. Also some of the reviewer´s questions can only be answered by additional experimental data as suggested.

Reviewer's Responses to Questions

**Part I - Summary**

Reviewer #1: This is a solid piece of work for the most part. There are some minor issues that need to be addressed.

In terms of novelty of the work, I was surprised by the lack of acknowledgement that this gene has been published on in very closely related T.brucei and there is no reference to this work.

There are far too many figures in the main manuscript and this should be reduced to the normal 8.

Reviewer #2: This manuscript describes an intriguing analysis of the second enzyme in the ethanolamine branch of the “Kennedy” pathway, ethanolamine phosphate cytidyltransferase (EPCT) within Leishmania major promastigote and amastigote lifecycle stages. The key takeaways are that

• the EPCT gene appears to be essential in both the promastigote and amastigote lifecycle stages and deletion of both EPCT chromosomal copies in promastigotes could only be achieved in the presence of a complementing episome.

• the episomal overexpression of EPCT caused a significant growth detect in both epct-/- promastigotes, as well as in WT, EPCT+/-, and epct-/- lesion derived amastigotes, and rendered these parasites more susceptible to nutrient starvation. Likewise, epct-/- promastigotes expressing EPCT from an episome were less tolerating of changes in pH and temperature.

• episomal overexpression of EPCT leads to dysfunction in the regulation of the “Kennedy” pathway and the synthesis of its downstream products.

• disruption of the ethanolamine metabolism by either deletion of one or both EPCT alleles or overexpression of EPCT from an episome affects the synthesis and surface expression of the GPI-anchored protein, GP63.

Overall, the systematic approach described in this manuscript makes a compelling case for EPCT as an important new drug target for future exploration in Leishmania. However, while the majority of the experiments described in the manuscript were sophisticated and well-executed, and the experimental outcomes convincing, there were a few areas where the manuscript could be improved, which I have detailed below.

Reviewer #3: In this study, the authors investigate the functional importance of an enzyme involved in lipid synthesis, Ethanolaminephosphate cytidyltransferase (EPCT), in Leishmania donovani promastigotes and amastigotes. This enzyme catalyzes the condensation of CTP with Ethanolamine phosphate to generate CDP-Ethanolamine, which is then used to generate both major sub-classes of ethanolamine phospholipids: plasmenylethanolamines (PME; major type in Leishmania) and diacyl-phosphatidylethanolamines (diacyl-PE). The authors generated heterozygote and homozygote null EPCT mutants complemented by an episomal EPCT copy that could be controlled by either positive or negative antibiotic selection. The authors then examined several phenotypic aspects of the mutants: (1) in vitro and in vivo growth of the mutants and their ability to lose the episomal copy; (2) EPCT activity, localization, and expression; (3) lipidomic changes; (4) downstream effects on lipid-anchored cell surface markers; and (5) impact on stress response.

One major finding of the work is that EPCT is essential in both promastigotes and amastigotes, based on the inability to generate homozygous nulls unless complemented with an episomal copy that cannot be lost, even under negative selection. A second major finding is that over-expression of EPCT is deleterious to the parasite, showing negative impacts on promastigote growth in culture and amastigote growth in mice (less virulent), reduction in PME and inositolphosphorylcholine (IPC) lipid species, reduction in GP63 surface expression, and impaired stress tolerance (starvation, acid, and heat). These major findings are clear and well-supported by the data and included controls.

One major strength of these findings is that they highlight the indispensable importance of EPCT to the parasite, even in the mammalian amastigote stage. This is in contrast to current thinking, based on results with other lipid synthesis mutants, that the amastigotes are less dependent upon their own lipid synthesis because they grow more slowly and can instead rely upon uptake of host lipids and lipid precursors. The results in this manuscript show that EPCT activity is still needed in the amastigote stage, implying that host resources or their acquisition is insufficient. A second strength is the very clear implication that too much EPCT activity is broadly deleterious for Leishmania, implying that a balanced tuning of lipid synthesis pathways is essential.

The manuscript does have some moderate weaknesses in its current form, which are summarized here and detailed below. The methods used for the studies shown in Fig. 4 and Fig. S5 are not described in sufficient detail for this reviewer to evaluate the data and conclusions. There is an inconsistency in the methods vs. the results with the generation of the null mutants. The potentially mixed effects of toxicity from ganciclovir treatment with toxicity from EPCT over-expression are not adequately addressed. Some speculation of effect on GP63 expression is needed. Finally, the results of the EPCT RNA expression data are confusing and not adequately addressed by the authors.

Overall, the manuscript has many strengths and represents a generally careful and thorough evaluation of the functional importance of EPCT in Leishmania. The key observations are well-supported and reveal that there is something special about the PE synthesis pathway, and more specifically, EPCT, which is needed in amastigote form and thus, constitutes an exciting new candidate drug target. So the work should be of interest to those working on drug development in eukaryotic pathogens. This research also will be of great interest to others focused more generally in lipid synthesis in kinetoplastids and other protozoan pathogens.

**Part II – Major Issues: Key Experiments Required for Acceptance**

Reviewer #1: There are four experimental issues that should be addressed.

1) in fig 2Athe authors use over expressing EPCT in the Leish background to show for activity. This is done by the level of the upper "CDP-Etn" band. Granted we see a more intense ban in the OE cell lines, but there is presence of this band in the buffer only (which is very concerning" and also the boiled (so dentures protein) controls, why would this be so. I would like to suggest that may be the higher band is not CDP-EthN, but may be Etn that has no P , in other words it has been dephosphorylkated. This could arise by the radioactive Etn-P partially degraded or phosphatase activity present, which there will be lots of in the lysate. Show me on a TLC where EthN runs rot be certain.

Fig 2B shows the anti-GFP protein bands which show several in the middle lane... thus when this is translated to the IF, then not only is the full length fusion showing up but also the truncated versions.. all of which are being visualised in the IF images. Localisations in cells normally requires a sub-cellular fractionation and IF, so the authors need to do a simple sub-cellular fraction western, then at least then they can be certain that the full length GFP +ve band is in the cytsol. also why do DAPI staining?

The authors clearly show that the levels of certain lipid species are decreased both in the cells over expressing or under expressing the EPCT. So the authors look at PE and PC, but also PI and IPC species, however the decrease in lipid content by these species, must be compensated for by another lid, which the authors have not shown, please show the full survey scans.. what else is going on? The cells obviously divide at a reasonable rate, so is their morphology or cell volume changed.. we do see any images of these modified cell. In the T.brucei work clear shows the cell volume has decreased, as well as the mitochondria... which the authors here do not even consider investigate with mitotracker, despite PE being so important for the Mito.

Reviewer #2: Major issues to be addressed:

• It’s unclear how the levels of GP63 were quantified for the different cell lines described? It appears that the values were derived from the immunofluorescence images captured. I have reservations about the accuracy of this approach, since I imagine it can be affected by any number of things including changes in the sample processing, gain and exposure settings, differences in the focal plane. At the very least, it would have been helpful to provide some indication for the number of cells enumerated and included another fluorescence marker (e.g., anti-tubulin) as a normalization control for the image analysis. A better and more standard approach would have been to include a quantitative western analysis using the anti-GP63 antisera.

Reviewer #3: Specific comments:

1. Fig 4 (Panels C-F) and Fig. S5 (Panel A) – Post-Sorting Recovery of High-GFP population. The methods and interpretation of this experiment are unclear. The GFP-low population in D (epct– + pXNG4-EPCT + GCV) was isolated by cell sorting then clones isolated by serial dilution. The population immediately after isolation is not shown, so quality of sorting is not able to be evaluated. Also it is not clear how the cells were propagated after sorting & cloning. During clone outgrowth, was GCV absent, was SAT present, or were all antibiotics omitted? Not clear how conclusion on Line 208 “GFP-low cells were only viable in the presence of GFP-high cells” is reached? What is this based on? Cannot follow that conclusion’s logic here. To me, this just seems like the recovery of the “high-GFP” population just reflects removal of GCV-negative selection on the episomal copy, and subsequent expansion of episomal copy number. qPCR of episomal copy number was performed (Fig. S5A), but it is not clear what time point after clonal isolation and outgrowth and in what culturing conditions (presence of antibiotics) were the samples taken for qPCR analysis. Was this immediately after cell sorting out the low GFP population?

2. Ganciclovir (GCV) treatment is toxic to cells expressing Thymidylate Kinase (TK) on the episome, which serves as a negative selection for the episomal EPCT copy. However, cells that cannot lose the episomal EPCT are potentially suffering from TWO effects that are happening at the same time: toxicity from GCV and toxicity from EPCT over-expression. I would like to see a little more discussion of this, and can your data shed any light on the relative contribution of GCV toxicity vs. EPCT over-expression might be having to the impaired growth you’re seeing in Fig. S5B and Fig. 5. I think there are some clues in your data that GCV toxicity might be contributing more to impaired growth in vivo in amastigotes than in Promastigotes, cross-correlating episome copy number vs. EPCT mRNA levels.

3. In addition to the point above, I’d like to see some attempt in the Discussion to consider the phenotypes in light of the episome copy number and EPCT mRNA data. Particularly as the Fig. 6C EPCT mRNA expression data is surprising, showing that GCV treatment increased EPCT mRNA levels in two of the EPCT over-expressing lines, when I would have expected it to decrease due to negative selection on the episome. The data is all a little puzzling as the in vivo lesion progression, the EPCT mRNA, and the episomal copy numbers don’t correlate very well.

4. In general the EPCT heterozygotes (+/–) had almost no phenotype in the various assays you did – indeed one copy seemed sufficient. So I was surprised that GP63 surface staining by immunofluorescence was as affected in the +/– as in the three over-expression mutants. Why do you think GP63 expression was so sensitive to EPCT haploinsufficiency?

5. Comment/Thought - One possibility for the effect of EPCT over-expression is that it could be exerting its effect through depletion of cellular CTP, as well as the changes you saw in PE and IPC species.

**Part III – Minor Issues: Editorial and Data Presentation Modifications**

Reviewer #1: I think most of fig 3 can be moved to supplementary, as can the quantification of the mRNA/copies per cell bar charts.

In fig 1 .. the pathway schematic they show the 1-alkyl-2-acylPE going to -alkenyl-acyl-PE, is this really true, have theu authors knowledge of a Desaturase that works solely on the 1-alkyl-2-acylPE with no suchance of 1-alkenyl-2-acyl being the direct acceptor to the EPT?

Following on from this are the authors certain there is no 1-alkyl-2-acyl PC, hence the SAM methyl transferases only work on diacyl species?

Reviewer #2: Minor comments for overall manuscript improvement:

• Figures S3 and S4: It’s unclear whether both of the heterozygotes shown were derived from the BSD gene replacement construct? The text (Lns 169-170) indicates this is the case. Did the authors try this the other way around? In other words, did they verify that the other gene replacement construct integrates appropriately into the chromosome? It’s not really a necessary control since the data with the epct-/- complemented cell line that is dependent on the presence of the episome implies that both gene copies were deleted, however, I am surprised that the authors didn’t provide Southern data showing the loss of both chromosomal alles.

• Figure S5b: I assume from the figure that just the growth point at day 2 was significantly different between the epct-/- complemented cell line and the other cell lines? What were the significance values for the other data points? I’m also curious why the cell lines appear to have different starting densities at day 0?

• While the data in Figure 5A was highly compelling, I found the text describing this figure to be less so, mainly due to the fact that the effect of EPCT overexpression wasn’t discussed alongside the other data. I think the two sections “EPCT is indispensable for L. major amastigotes” and “EPCT overexpression leads to significantly attenuated virulence in mice” could be combined and shortened. This would serve to focus the attention on the key takeaways for this figure.

• For Figure 6C, I noticed that EPCT mRNA levels were increased significantly for WT and epct-/- cell lines containing the pXNG4-EPCT episome in the presence of GCV in comparison to PBS control, which seems a little surprising. Perhaps the authors can comment a little more on this observation in the text.

• Figures 7-9 contain a large amount of data (changes in the level of several PE, PC, and PI metabolites for several different cell lines). I suggest moving this data to a table to make it easier to digest (though I’d focus the table on only the major differences). Figures 7 – 9 could still be included, but as supplemental data.

• The starvation data is intriguing and I wonder if this phenotype might be related to defects in autophagic vesicle formation. Perhaps the authors could speculate on this possibility? Finally, I’m curious whether the authors have considered exploring the susceptibility of the complemented cell lines (WT, EPCT+/-, and epct-/-) to antileishmanial drugs? On the one had these cells seem to be more susceptible to stressors such as nutrient starvation, and pH and heat shock, but

• Figure 2B and C: GFP-EPCT is clearly overexpressed in these cells, but from the western analysis it appears that there are also some likely degradation products, possibly resulting in GFP protein alone. I’m wondering how this might affect the assertion of GFP-EPCT being a cytosolic protein? The authors mention that the two enzymes immediately downstream from EPCT are localized to the ER. What is known about the enzyme immediately upstream of EPCT? Perhaps the authors can speculate on whether the cytosolic location makes sense for EPCT in the context of the other proteins?

• In the discussion, Lns 330-331, the phrase “it is time to recognize EPCT as the most important enzyme of the “Kennedy” pathway needs rephrasing to be less subjective.

• In the discussion, Ln 359, the authors discuss substrate inhibition of the downstream enzymes CEPT and EPT as a plausible cause of decreased diacyl PE and PME production. I wonder what is known about the association of these different activities in other cells, specifically, whether they can be found in complex? In this context, overexpression of EPCT might also disrupt the stoichiometry of the other components of the “Kennedy” pathway, leading to decreased production of the downstream products.

Reviewer #3: 6. In description of constructing the episome-assisted KOs (Lines 174-185), the text says gene replacement with BSD cassette was first, followed by episome introduction, then deletion of second allele with PAC cassette. But in Fig. 3A and using Fig. S4 scheme, it looks like PAC was first replacement then BSD. In Fig. 3A – the heterozygote shows the PAC band and the WT band (lane 2), implying it was the first gene replacement.

7. In Discussion, some care should be made about concluding localization of EPCT to cytoplasm using an over-expression construct (Fig. 2C). It could be the higher level of expression of EPCT could “swamp” out the localization mechanism, leading to overall cytoplasmic pattern. It’s consistent with cytoplasmic localization (and localization of other EPCT enzymes in other organisms), but localization at WT levels of expression need to be done to confirm localization.

8. In Fig. 10, it is not indicated in the Materials and Methods how the quantitation in panel C was done. Is this quantitation from a western blot or from the immunofluorescence? This info should be added to the M&M.

PLOS authors have the option to publish the peer review history of their article (what does this mean?). If published, this will include your full peer review and any attached files.

Reviewer #1: No

Reviewer #2: No

Reviewer #3: **Yes: **Kimberly Paul
---

## [Decision Letter · Decision Letter 1]

17 Jun 2023

Dear Dr. Zhang,

Thank you very much for submitting your manuscript "Ethanolaminephosphate cytidyltransferase is essential for survival, lipid homeostasis and stress tolerance in Leishmania major" for consideration at PLOS Pathogens. As with all papers reviewed by the journal, your manuscript was reviewed by members of the editorial board and by several independent reviewers. The reviewers appreciated the attention to an important topic. Based on the reviews, we are likely to accept this manuscript for publication, providing that you modify the manuscript according to the review recommendations as indicated below.

Reviewers of the revised manuscript acknowledge that additional data, text additions and corrections have significantly improved the work. However, a number of minor points remain to be corrected as indicated in their specific comments. In addition, we agree with reviewer 1 that the IF in Fig. 2D does not prove cytoplasmic localization. The resolution of the images is relatively low and absence of a specific subcellular localization of a tagged protein does not positively argue for cytoplasmic localization. A simple cell fractionation (100.000 g supernatant) as suggested by reviewer 1 would be much more informative and should be provided. Fig. 3 should be moved to supplement, also to conform with journal standards.

Sincerely,

Michael Boshart, M.D.

Academic Editor

PLOS Pathogens

Margaret Phillips

Section Editor

PLOS Pathogens

Kasturi Haldar

Editor-in-Chief

PLOS Pathogens

orcid.org/0000-0001-5065-158X

Michael Malim

Editor-in-Chief

PLOS Pathogens

orcid.org/0000-0002-7699-2064

Reviewer Comments (if any, and for reference):

Reviewer's Responses to Questions

**Part I - Summary**

Reviewer #2: This manuscript takes a systematic approach to dissect the function and significance of EPCT for promastigote growth and amastigote virulence. The key findings suggest that contrary to the prevailing theory that Leishmania amastigotes rely solely on the uptake of host lipid to fulfill their needs, EPCT is an essential enzyme with likely a regulatory role in both stages of the parasite. The revised manuscript is much improved in both layout and its experimental detail, and I consider the experiments to be well-executed and the conclusions drawn generally to be supported by the results.

Reviewer #3: In this study, the authors investigate the functional importance of an enzyme involved in lipid synthesis, Ethanolaminephosphate cytidyltransferase (EPCT), in Leishmania donovani. The authors present evidence that EPCT is required in both the promastigote stages as well as in vivo amastigotes. The requirement in amastigotes is novel, as prior work on other lipid synthesis genes suggest amastigotes rely primarily on host lipids rather than de novo synthesis. The other interesting and novel finding is that over-expression of EPCT is deleterious as well, perturbing PE and PI species, causing reduced virulence, and reduced tolerance to stress. Overall, this work is novel and interesting.

The authors largely addressed all the reviewers' comments. The revision included an expanded discussion, more experimental details, and additional experiments. The manuscript is much improved in clarity and impact.

**Part II – Major Issues: Key Experiments Required for Acceptance**

Reviewer #2: None noted

Reviewer #3: None

**Part III – Minor Issues: Editorial and Data Presentation Modifications**

Reviewer #2: There are just a few very minor edits that I would suggest to improve the clarity of the manuscript.

1. For lns 284 and 410, where the authors indicate that the effects observed are due to a wearing down of GCV, I'd suggest changing this language to "may be due to the level of GCV likely being below the minimum effective concentration".

2. I believe that for ln 327 the figure the authors refer to should be 9D (not 9C) and for ln 329 Fig. 9 D-F should be Fig. 9C and F.

Reviewer #3: Minor edits:

Line 79 - should read "...(EtN-P) is generated from the sphingoid..."

Line 327 - This refers to Fig. 9D not 9C.

Line 329 - Refers to Fig. 9C,E,F not Fig. 9D-F

Line 340 - "combing" should be "combining"

Line 699 - Refers to Fig. 3 not Fig. 2

Line 738-739 - Statement about the structures of major PC species illustrated should be deleted. No structures are in the figure.

Line 742 and Fig. S11 - It looks like in the spectra that the indicated internal standard was not set at 100% relative intensity. Instead, it looks like d16:1/18:0-IPC (m/z 778.6) is set to 100% relative intensity and the other peaks normalized to that species. Maybe that IPC species was the internal standard?

Figure 8B - not sure what band(s) correspond(s) to LPG? Which band or set of bands is being quantified to generate the data in Fig. 8C?

PLOS authors have the option to publish the peer review history of their article (what does this mean?). If published, this will include your full peer review and any attached files.

Reviewer #2: No

Reviewer #3: **Yes: **Kimberly Paul

Figure Files:

Data Requirements:

Reproducibility:

References:

---

## [Editor Report · Decision Letter 2]

12 Jul 2023

Dear Dr. Zhang,

We are pleased to inform you that your manuscript 'Ethanolaminephosphate cytidyltransferase is essential for survival, lipid homeostasis and stress tolerance in Leishmania major' has been provisionally accepted for publication in PLOS Pathogens.

Best regards,

Michael Boshart, M.D.

Academic Editor

PLOS Pathogens

Margaret Phillips

Section Editor

PLOS Pathogens

Kasturi Haldar

Editor-in-Chief

PLOS Pathogens

orcid.org/0000-0001-5065-158X

Michael Malim

Editor-in-Chief

PLOS Pathogens

orcid.org/0000-0002-7699-2064

The authers have successfully performed the addition cell fractionation experiments that supports cyctplasmic localization and responded to the reviewer comments. Their response to the following requests for clarification of reviewer 3 has not yet been included in a modified legend for figures 8 and S11. This can be easily done.

start citation-Line 742 and Fig. S11 - It looks like in the spectra that the indicated internal standard was not set at

100% relative intensity. Instead, it looks like d16:1/18:0-IPC (m/z 778.6) is set to 100% relative intensity

and the other peaks normalized to that species. Maybe that IPC species was the internal standard?

In mass spectrometry chromatogram, the most abundant peak is automatically set as 100%. In Fig. S11,

the abundance of d16:1/18:0-IPC (m/z 778.6) is higher than that of the internal standard 16:0/16:0-PI

(m/z 809.6). Quantitation was still done using the internal standard.

Figure 8B - not sure what band(s) correspond(s) to LPG? Which band or set of bands is being quantified

to generate the data in Fig. 8C?

LPG is a heterologous in glycan chain length and side chain modification (PMID: 1444269). The antibody

WIC79.3 detects the side chain Gal residues and a smear for LPG (PMID: 6184664). The area between-end citation

15-100 KD was used to quantify LPG.
---

## [Editor Report · Acceptance letter]

22 Jul 2023

Dear Dr. Zhang,

We are delighted to inform you that your manuscript, "Ethanolaminephosphate cytidylyltransferase is essential for survival, lipid homeostasis and stress tolerance in Leishmania major," has been formally accepted for publication in PLOS Pathogens.

Best regards,

Kasturi Haldar

Editor-in-Chief

PLOS Pathogens

orcid.org/0000-0001-5065-158X

Michael Malim

Editor-in-Chief

PLOS Pathogens

orcid.org/0000-0002-7699-2064